# Molecular features of the UNC-45 chaperone critical for binding and folding muscle myosin

Doris Hellerschmied[1,5,6]*, Anita Lehner[2,6], Nina Franicevic[1,6], Renato Arnese [1], Chloe Johnson [3], Antonia Vogel[1], Anton Meinhart[1], Robert Kurzbauer[1], Luiza Deszcz[1], Linn Gazda[1], Michael Geeves [3] & Tim Clausen [1,4]*

Myosin is a motor protein that is essential for a variety of processes ranging from intracellular transport to muscle contraction. Folding and assembly of myosin relies on a specific chaperone, UNC-45. To address its substrate-targeting mechanism, we reconstitute the interplay between *Caenorhabditis elegans* UNC-45 and muscle myosin MHC-B in insect cells. In addition to providing a cellular chaperone assay, the established system enabled us to produce large amounts of functional muscle myosin, as evidenced by a biochemical and structural characterization, and to directly monitor substrate binding to UNC-45. Data from in vitro and cellular chaperone assays, together with crystal structures of binding-deficient UNC-45 mutants, highlight the importance of utilizing a flexible myosin-binding domain. This so-called UCS domain can adopt discrete conformations to efficiently bind and fold substrate. Moreover, our data uncover the molecular basis of *temperature-sensitive* UNC-45 mutations underlying one of the most prominent motility defects in *C. elegans*.

[1] Research Institute of Molecular Pathology, Vienna BioCenter, Vienna, Austria. [2] Vienna BioCenter Facilities, Vienna, Austria. [3] School of Biosciences, University of Kent, Canterbury, UK. [4] Medical University Vienna, Vienna, Austria. [5]Present address: Faculty of Biology, Center of Medical Biotechnology, University Duisburg-Essen, Essen, Germany. [6]These authors contributed equally: Doris Hellerschmied, Anita Lehner, Nina Franicevic. *email: doris. hellerschmied@uni-due.de; tim.clausen@imp.ac.at

Myosins are cytoskeletal, molecular motors promoting a variety of mechano-chemical processes in the cell[1]. The myosin superfamily contains about 35 subtypes, among which the class II muscle proteins (myosin II) are the arguably most prominent member driving muscle contraction[2]. Myosin II molecules found in skeletal and cardiac muscles dimerize via their C-terminal coiled-coil domain and associate with essential and regulatory light-chains to yield the basic hexameric myosin complex. These hexamers further assemble into myosin thick filaments, from where the N-terminal myosin ATPase domains project to interact with the adjacent actin (thin) filaments. Establishing and maintaining the intricate myosin–actin interplay, occurring at the interface of the two differently organized muscle filaments, is a great challenge for the cellular chaperone machinery. Therefore, muscle cells express a vast number of specialized folding and assembly factors that control the expression, folding, assembly and interplay of actin and myosin molecules[3–7]. Failure of muscle filament assembly or maintenance leads to severe myopathies[8,9].

Among the chaperones expressed in muscle cells, UNC-45 appears to be the key component of the myosin assembly machinery. The UNC-45 protein was initially identified in *C. elegans*, where site-specific *temperature-sensitive (ts)* mutants revealed the importance of the chaperone for myosin function[10–13]. Subsequent studies in *C. elegans*, Xenopus, Zebrafish, and Drosophila confirmed the role of UNC-45 as a myosin-specific chaperone, promoting the folding of the myosin ATPase domain as well as coordinating the assembly of thick filaments during muscle development[14–19]. In higher organisms, UNC-45 is particularly important for the maturation of myosin II in muscle cells[20–22] and, together with Hsp90 (heat shock protein 90) and UFD-2 (ubiquitin fusion degradation 2), maintaining the functionality of myosin filaments during stress situations[23,24]. Mechanistic insight into the chaperone function of UNC-45 comes from structural studies revealing its 3-domain architecture[19,25]. UNC-45 is composed of an N-terminal TPR (tetratricopeptide repeat) domain mediating the interaction with Hsp70/Hsp90, an elongated UCS (UNC-45/Cro1/She4p) domain at the C-terminus providing a myosin-binding site[14,19], and a central domain aligning the two functional TPR and UCS units to each other. Further analyses of UNC-45 from *C. elegans* revealed that the chaperone can form a linear protein chain, which constitutes a myosin assembly line licensing Hsp70 and Hsp90 to act in a defined periodicity on myosin heads protruding from the myofilaments[19,26].

To address the myosin targeting mechanism of UNC-45, we reconstitute the chaperone-substrate interplay both in vitro and in vivo. Using insect cells as host system, we monitor the interaction between the *C. elegans* UNC-45 and MHC-B (myosin II heavy chain isoform B, also known as UNC-54) in a cellular context. Notably, co-expression of UNC-45 allowed production of fully functional MHC-B in large amounts, yielding about 15 mg muscle myosin per liter culture. The recombinant myosin was also key to address the basic mechanistic properties of the UNC-45 chaperone, revealing for example the molecular basis of *ts* motility defects of mutant worms harboring point-specific UNC-45 mutations. Our data show that these *ts* mutations affect the myosin-binding capability of UNC-45 rather than its protein stability.

## Results

**In vivo reconstitution of the UNC-45/myosin interplay**. An inherent problem in characterizing the substrate-targeting mechanism of the UNC-45 chaperone comprises the unavailability of the cognate substrate, muscle myosin II. We thus aimed

to establish an orthogonal in vivo assay to monitor the activity of myosin-specific chaperones. To this end, we used the motor domain of *C. elegans* body wall muscle myosin MHC-B as model system and co-expressed it with different *C. elegans* chaperones in insect cells (Fig. 1a). We first tested the production of an MHC-B muscle myosin variant comprising the motor domain (residues 1–790) in comparison to a non-muscle myosin motor (nematode NMY-2, residues 1–796). While the NMY-2 motor domain could be expressed in soluble form, even in the absence of any *C. elegans* helper chaperone (Fig. 1b), the expression of the MHC-B muscle myosin alone did not yield any soluble recombinant protein, a finding which is consistent with previous reports[21,27]. As it is known that the *C. elegans* chaperones UNC-45, HSP-1 (Hsp70) and DAF-21 (Hsp90) are critical for myosin folding and assembly[14,19,28], we next tested whether co-expression of these chaperones improves the production of the MHC-B motor domain in its soluble form. The experiments revealed that the *C. elegans* Hsp70 and Hsp90 had only a moderate effect in yielding soluble muscle myosin in insect cells. However, co-expressing UNC-45 strongly increased the amount of the MHC-B motor domain in the soluble fraction of the cell lysate (Fig. 1c). These data imply that the nematode UNC-45 can team up with the insect cell chaperone machinery required for myosin folding, given that additional co-expression of the cognate partner chaperones Hsp70 and Hsp90 from *C. elegans* was not required to obtain soluble myosin (Fig. 1c). Indeed, we could pull-down endogenous Hsp70 and Hsp90 together with *C. elegans* UNC-45 from insect cell lysates (Fig. 1d). This interaction is abolished upon deletion of the UNC-45 TPR domain or mutating a key residue (K82E) in the Hsp70/90 binding groove, while deletion of the UCS domain does not impact the interaction with the partner chaperones (Fig. 1d). Finally, when purifying MHC-B and testing its actin-induced ATPase activity, we did not observe major differences in the amount and functionality of the myosin motor co-expressed with UNC-45 alone or together with *C. elegans* Hsp70 and Hsp90 (Fig. 1e and Supplementary Fig. 1). Taken together, our findings suggest that the *C. elegans* UNC-45 is the most critical chaperone for making MHC-B muscle myosin.

**Analyzing the UNC-45/myosin interplay in insect cells**. Since the recombinant production of functional MHC-B myosin fully depends on UNC-45, the insect cell co-expression system can be used as a cellular assay to determine the effect of site-specific UNC-45 mutations on its de novo myosin folding activity. To this end, we analyzed functional UNC-45 mutations that abrogated interactions with partner chaperones or substrate, prevented formation of multimeric UNC-45 chains or led to the well-known *ts* phenotype in nematodes (Fig. 2a). To assess the folding state of MHC-B co-expressed with different UNC-45 mutants, we monitored its solubility in insect cell lysates (Supplementary Fig. 2a) and, in parallel, analyzed the purified proteins in vitro by analytical SEC and ATPase assays. Fitting to our insect cell data (Fig. 2b), the strongest difference in SEC runs was observed for MHC-B expressed either alone or in the presence of the UNC-45 wild-type (wt) chaperone (Fig. 2c). In the absence of UNC-45, the produced MHC-B is mostly misfolded, eluting at the void volume of the SEC column (fractions 1–4, Fig. 2c, d). Contrary, in the presence of UNC-45, most of the MHC-B protein elutes in a pronounced peak that corresponds to the folded myosin motor (fractions 5–9, Fig. 2c, d). While the SEC runs allowed distinguishing between aggregated and folded protein, the ATPase assays confirmed that the isolated MHC-B motors, eluting in the folded peak, are enzymatically active, exhibiting comparable, actin-inducible ATPase activity (Supplementary Fig. 1). However, it should be noted that the SEC profiles strongly varied between

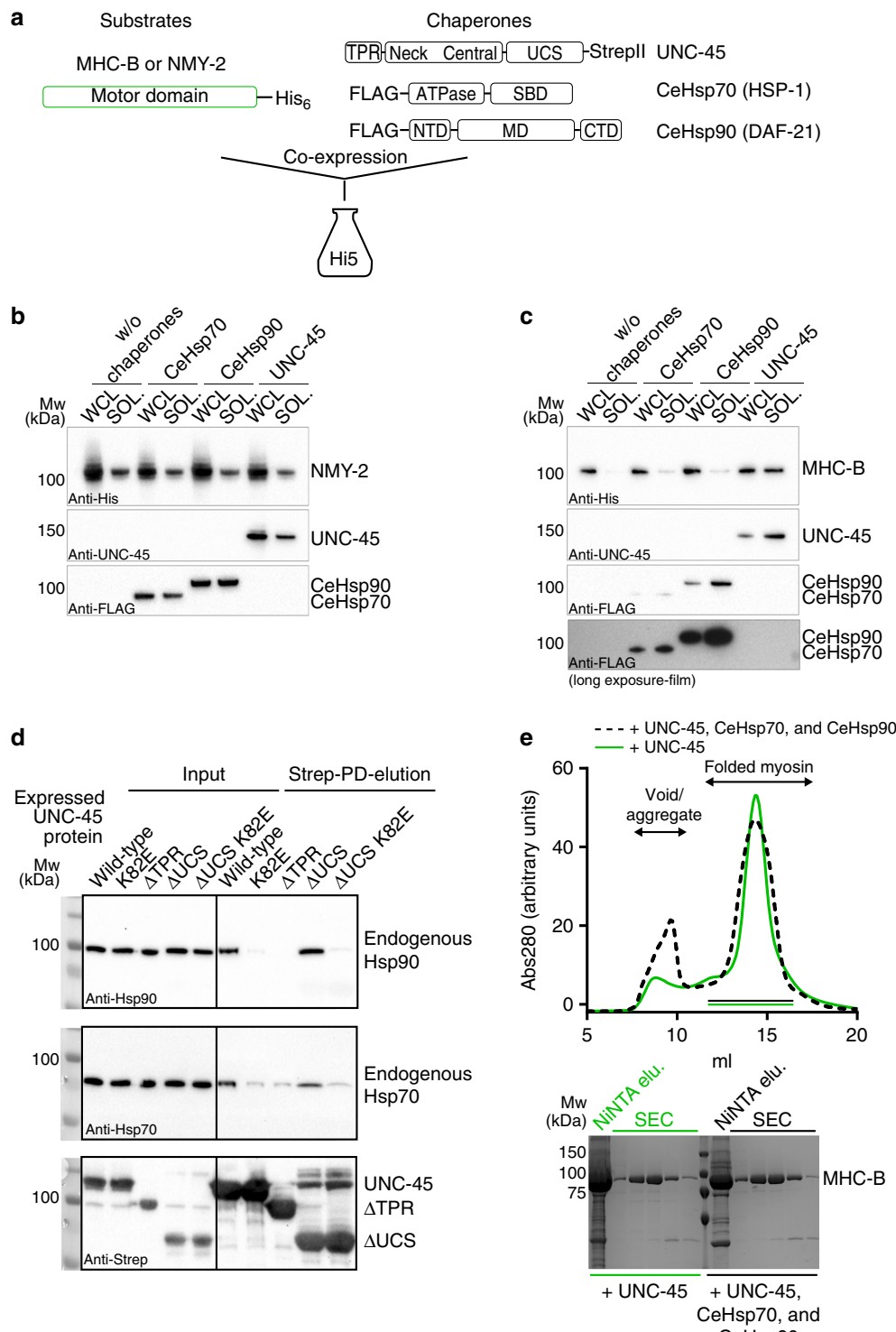

**Fig. 1** Producing functional myosin in insect cells. **a** Overview of constructs used for expression studies in insect cells. Western blot analysis of co-expression of NMY-2 (**b**) or MHC-B (**c**) with indicated chaperones in Hi5 insect cells. Whole cell lysates (WCL) and soluble fractions (SOL) are shown. 1/3rd of the WCL was applied with respect to the SOL fraction. (SBD–substrate binding domain, NBD–nucleotide binding domain, NTD–N-terminal domain, MD–middle domain, CTD–C-terminal domain). **d** Pull-down (PD) of Strep-tagged UNC-45 constructs from insect cells. Input samples and elutions were probed with Hsp70, Hsp90, and Strep-tag antibodies. **e** *Upper panel*: SEC traces of MHC-B purified from co-expression with the indicated *C. elegans* chaperones. *Lower panel*: SDS-PAGE gels showing the elution after NiNTA and indicated SEC fractions

different samples, depending on the co-expressed UNC-45 cha-perone. We therefore used the SEC-purified amounts of mono-meric, folded MHC-B (Fig. 2d) together with the data from our cellular assay (Fig. 2b and Supplementary Fig. 2a) as a readout to assess the chaperone activity of the different UNC-45 forms.

We first studied two mutants deficient in myosin binding, namely a deletion mutant lacking the UCS domain (residues 1–521, ΔUCS)[14,24] and a variant carrying a point mutation (N758Y) in the myosin-binding canyon[19]. Deletion of the UCS domain had a strong impact on UNC-45 chaperone function, as

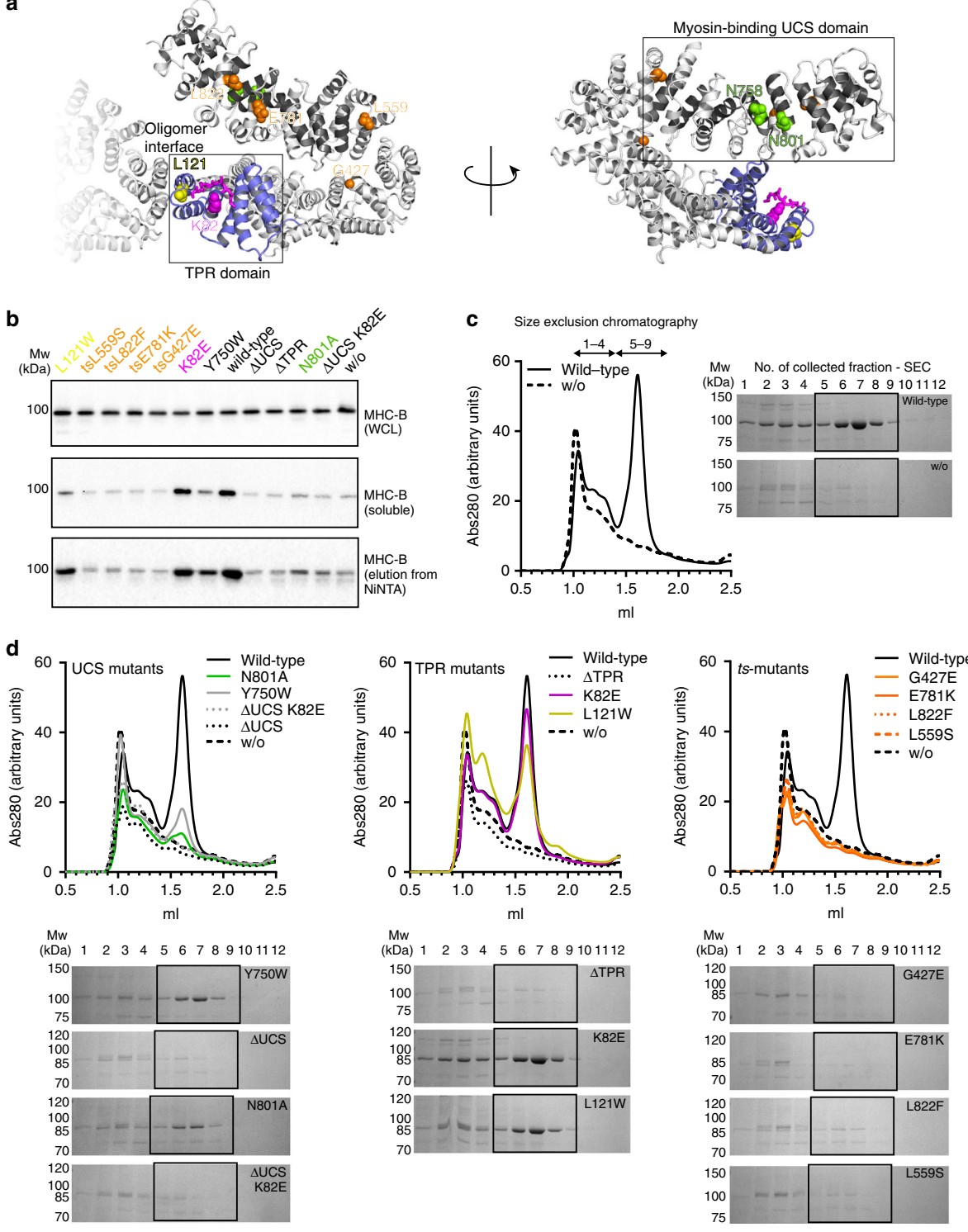

**Fig. 2** Effect of UNC-45 mutant proteins on myosin production in insect cells. **a** UNC-45 mutations that were analyzed in the chaperone assay are shown on the crystal structure of the wild-type protein (PDB code: 4i2z). **b** Western blot analysis of co-expression of MHC-B with indicated UNC-45 mutant proteins in insect cells. Whole cell lysates (WCL), soluble fractions and elutions after small-scale NiNTA purification are shown for MHC-B. **c** SEC traces and corresponding SDS-PAGE gels for MHC-B purified from insect cells with and without co-expression of UNC-45. Fractions 1–4 cover the void/aggregate peak, while fractions 5–9 correspond to the elution volume of monomeric functional myosin. **d** Same analysis as in **c** for MHC-B co-expressed with UNC-45 UCS domain mutants, TPR domain mutants and *ts*-mutants

seen by the reduced levels of soluble MHC-B in the insect cell co-expression system (Fig. 2b and Supplementary Fig. 2a). The site-specific N758Y mutant protein itself was not highly expressed in insect cells, likely because the mutation destabilizes the UCS

domain as observed for other point mutations bordering the myosin-binding groove[24]. We therefore excluded the N758Y mutant from our analysis. Instead, we used the cellular chaperone assay to look for a new site-specific UCS mutation having an

impaired myosin folding activity. A comparative analysis with beta-catenin, which is an armadillo (ARM) repeat protein structurally related to the UCS domain, suggested that Asn801 of UNC-45 may directly interact with myosin (Supplementary Fig. 3). The N801A variant is stable in insect cells, however compared to the wt protein, co-expression of the UCS mutant yielded only small amounts of soluble myosin (Fig. 2b and Supplementary Fig. 2a). Moreover, we observed only a minor peak of the folded MHC-B in the SEC analysis (Fig. 2d), indicating that the N801A mutation strongly impaired the myosin-folding function of UNC-45. Finally, we explored the effect of the UNC-45 mutation Y750W, which affects a residue located in the periphery of the myosin-binding canyon and did not show a clear phenotype when expressed in *C. elegans*[19]. While Y750W supported the production of soluble myosin in the cell (Supplementary Fig. 2a), the yield of functional MHC-B that could be purified is smaller than for wt UNC-45 (Fig. 2d). These data suggest that even subtle structural changes in proximity to the UCS myosin-binding site affect UNC-45 chaperone function.

The interaction of UNC-45 with its partner chaperones Hsp70/Hsp90 is mediated via its TPR domain[14,19]. When we assayed the activity of the K82E mutant, which is impaired in Hsp70/90 binding (Fig. 1d, and ref. [19]), we observed that it yielded similar levels of soluble myosin as the co-expressed wt UNC-45 (Fig. 2b). Notably, the SEC peaks of the folded MHC-B motor domain purified from the two samples almost completely overlap (Fig. 2d), implying that the K82E mutant exhibits full chaperone activity. These data suggest that a direct interaction between the Hsp70/Hsp90 chaperones and the UNC-45 TPR domain is not required to fold the motor domain of myosin. Surprisingly, however, deleting the entire TPR domain fully abolished the chaperone function of UNC-45 (Fig. 2b, d). Accordingly, the TPR domain does not only serve as Hsp70/Hsp90 docking site, but accomplishes further critical functions.

The linear UNC-45 chaperone chain has been shown to be essential for myofilament formation and sarcomere integrity in worms[19]. To test the importance of oligomer formation for myosin folding, we analyzed UNC-45 L121W, a single-site mutant known to disrupt the interface stabilizing UNC-45 chains. As seen in the insect cell chaperone assay, the L121W mutation led to a decrease in the amount of soluble MHC-B (Supplementary Fig. 2a). However, in our SEC analysis, a significant amount of the soluble MHC-B elutes as folded protein (Fig. 2d), suggesting that UNC-45 oligomerization is not required for the maturation of the myosin motor domain.

In *C. elegans*, four distinct UNC-45 point mutations (G427E, L559S, E781K, L822F) have been identified that underlie the so-called uncoordinated (unc) *ts*-phenotype[10,11,13,29]. *Ts*-worms grown at the permissive temperature of 15–18 °C during their larval development show no defects in myosin thick filament formation, while *ts*-worms grown at the restrictive temperature of 25 °C exhibit motility defects and a distorted sarcomere organization[10,11]. Though known for decades, the mechanistic details underlying the *ts*-phenotype of UNC-45 have remained elusive. Strikingly, in our cellular chaperone assay, all *ts*-mutations strongly reduced the production of soluble myosin (Fig. 2b). Moreover, applying the minimal amounts of soluble protein to a SEC column did not recover any folded protein (Fig. 2d). To test whether this defect in chaperone function relates to a reduced stability of the *ts*-proteins themselves, we repeated the cellular folding assays at a lower temperature (18 °C) that still supported survival of insect cells and corresponds to the permissive temperature in growing *ts*-worms. Also, at 18 °C we observed the same solubility defect for myosin co-expressed with the *ts*-mutants (Supplementary Fig. 2b). These results suggest that the UNC-45 *ts*-mutants, which were expressed at the same level

as the wt protein, affect the chaperone function of UNC-45 rather than its stability.

**UNC-45 co-expression yields functional muscle myosin.** In addition to serving as a folding reporter in insect cells, we wanted to use the recombinant MHC-B motor domain as a substrate to characterize the UNC-45 chaperone in vitro. For this purpose, it was necessary to assess the structural and functional integrity of the produced muscle myosin in detail. We first tested the basal ATPase activity of purified, monomeric MHC-B in a stopped-flow setup by monitoring its tryptophan fluorescence (Fig. 3a). Upon ATP binding to the myosin motor domain Trp fluorescence is enhanced[30,31], and by monitoring its decay over time we determined an MHC-B ATPase rate of 0.073 s$^{-1}$ showing that the recombinant protein is active. We then explored the interaction of recombinant MHC-B with rabbit skeletal muscle F-actin in a co-sedimentation assay (Fig. 3b). In the absence of nucleotide, MHC-B interacts strongly with F-actin, as demonstrated by the large amounts of MHC-B seen in the pelleted fraction. However, when adding ATP, myosin was released from F-actin and detected in the supernatant, reflecting the nucleotide-dependent interplay between the two proteins (Fig. 3b). Next, the activation of MHC-B ATPase activity by F-actin was quantified in an NADH-coupled assay (Fig. 3c). Upon elevating the actin concentration, we observed a 6-fold increase of its basal ATPase activity, which inclined from 0.043 ATP s$^{-1}$ in the absence of actin to a $V_{max}$ of 0.257 ATP s$^{-1}$. Finally, and most importantly, the large amount of recombinant MHC-B produced enabled us to perform a comprehensive analysis of the interaction with actin during the ATPase cross-bridge cycle. To compare the kinetic parameters of MHC-B and human cardiac myosin, which served as our reference, we performed stopped-flow measurements exploiting the strong quenching of pyrene-labelled actin by MHC-B. Rapid mixing of an excess of ATP with pyrene-actin: myosin results in dissociating of the complex and a return of the fluorescence to the starting value (inset Fig. 3d). Measuring the amplitude of the ATP induced transient gives an estimate of the amount of actin:myosin complex formed at each myosin concentration. The resulting fit of the binding isotherm reveals an affinity for actin of 10.3 nM, which is comparable with the affinity of the human cardiac muscle myosin isoform for the same actin (10 nM) (Fig. 3d, Supplementary Table 1, and ref. [32]). Using similar stopped-flow experiments, we carried out a broad kinetic characterization of the MHC-B myosin motor, determining the rate constants of ATP and ADP binding to myosin and the actin: myosin complex, respectively. In conclusion, the kinetic data demonstrate that the MHC-B variant produced in insect cells is a fully functional myosin motor, which exhibits similar kinetic properties to other muscle myosins (Supplementary Table 1, Supplementary Fig. 4).

Finally, we crystallized MHC-B and determined the structure of the ADP-bound state to a resolution of 1.9 Å (Supplementary Table 2). The motor domain adopts the canonical myosin ATPase fold being comprised of the N-terminal, the central (upper and lower 50 kDa region) and a C-terminal domain, with all residues (except loops 198–210 and 633–650) being well-defined by electron density. Notably, a DALI search[33] for related protein structures in the Protein Data Bank identified the human beta-cardiac muscle myosin (PDB 4db1) among the closest structural homologs of the determined MHC-B structure. The two motor domains are 59% identical and 72% similar (Supplementary Fig. 5), and could be superimposed with an RSMD of 2.1 Å for 696 Cα atoms (Fig. 3e). Indeed, the nucleotide binding site of MHC-B is virtually identical to that of the human beta cardiac myosin, with the sidechains that coordinate ADP present in

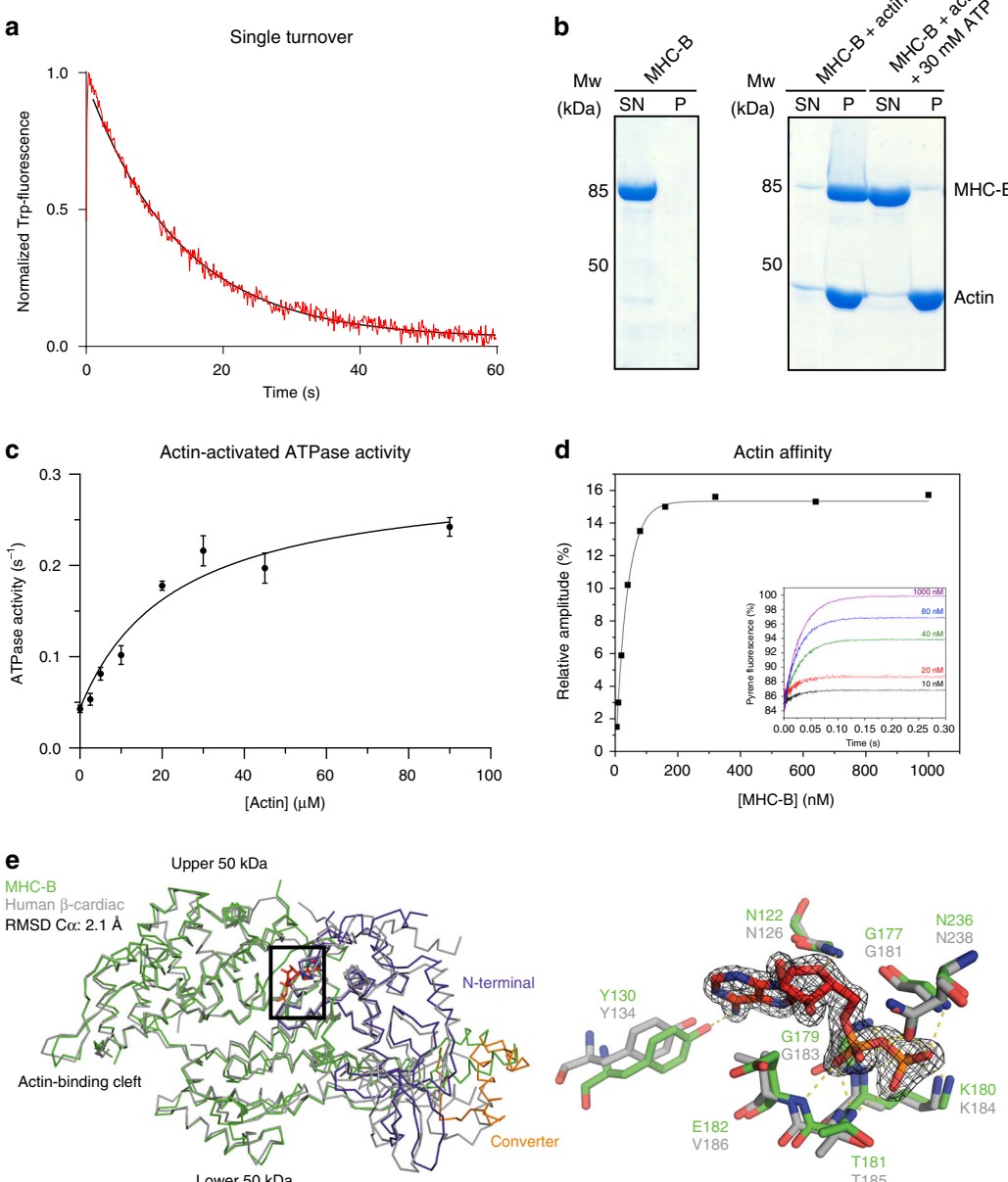

**Fig. 3** Characterization of functional MHC-B. **a** Single ATP turnover experiment monitoring Trp-fluorescence decay ($K_{obs} = 0.073\,s^{-1}$). **b** F-actin co-sedimentation assay. In the presence of F-actin, MHC-B is detected in the pellet fraction (P) but is released upon addition of ATP and found in the supernatant (SN). **c** F-actin-stimulated ATPase activity of MHC-B, yielding $V_{max} = 0.257\,s^{-1}$ and $K_M = 23.1\,\mu M$. Error bars represent the s.e.m., of two biological replicates (MHC-B from different purifications), for which three technical replicates have been carried out each. **d** Stopped-flow analysis measuring dissociation of the actin:MHC-B complex. The inset illustrates example traces of the increase in pyrene fluorescence when 20 μM ATP was mixed with 50 nM pyrene-labelled actin and varying concentrations of MHC-B (5–1000 nM). The increase in fluorescence can be described by an exponential function in which the rate constant is proportional to the ATP concentration and the amplitude of the fluorescence change is proportional to the fractional saturation of actin with MHC-B. The actin affinity derived by this experiment was 10.3 ± 3.7 nM (average of three independent measurements). **e** Left panel: Structural alignment of ADP(red)-bound MHC-B (50 kDa domain in green, N-terminal domain in lilac, converter in orange) and human beta-cardiac myosin (PDB code: 4db1, gray). Right panel: Close-up view of the nucleotide-binding pocket of MHC-B in the ADP-bound state, overlaid with corresponding active site residues of the cardiac myosin. The ADP molecule (red) is shown together with the 2Fo-Fc omit electron density, contoured at 1.5σ

almost identical position (Fig. 3e). In contrast to the human protein, which has to be produced in a special mouse cell line requiring time- and resource-demanding procedures, the biochemical and structural analysis of MHC-B was performed with a recombinant protein that can be produced in large amounts in insect cells (15 mg myosin per liter cell culture) and is easily

accessible to genetic manipulations. Considering the high yields and the overall similarity to mammalian muscle myosins, including the human beta-cardiac variant, the recombinant MHC-B should represent a valuable model system to address the assembly, function and regulation of muscle myosins in basic and in medical research.

**In vitro interaction of UNC-45 with damaged myosin**. To study how UNC-45 interacts with myosin molecules in vitro we used the recombinant MHC-B motor domain. Previous reports, in which non-cognate myosin variants and unrelated model substrates were used as chaperone substrates, implied that UNC-45 preferentially targets aberrant protein molecules[34,35]. However, the nature of the non-native myosin species targeted by UNC-45 —either during de novo folding or when rescuing damaged myosin—is not known. Thus, the targeted state of myosin (e.g. partially unfolded, misfolded or aggregated protein) and the molecular details of forming cognate UNC-45/myosin complexes need to be resolved. Instead of applying the rather indirect aggregate prevention assay, we monitored formation of the UNC-45/myosin complex at two different temperatures, 4 and 27 °C, in a size-exclusion chromatography (SEC) based assay. The 4 °C condition served as control temperature, preserving the native state of the myosin substrate (Fig. 4a). The higher temperature, 27 °C, reflects a proteotoxic stress condition, as implied by the fact that *C. elegans* worms exhibit signs of heat-shock and sterility at this temperature[36].

When evaluating the stability of the individual proteins at the elevated temperature (27 °C), we observed that myosin was destabilized and eluted in the void volume of the SEC column as expected for aberrant proteins. In contrast, all tested UNC-45 variants were stably folded, eluting at the same retention volume as the native chaperone at 4 °C (Fig. 4a, c). When myosin and UNC-45 were co-incubated at 27 °C, a prominent fraction of UNC-45 was present in complex with unfolded myosin eluting at higher molecular weight (HMW, Fig. 4a). Interestingly, when myosin and UNC-45 were separately incubated and mixed before loading, less UNC-45 was present in the corresponding HMW complex with myosin (Fig. 4a). This suggests that during myosin denaturation, a certain un-folding intermediate is formed that is preferentially targeted by UNC-45. To validate that the observed interaction is specific, we used bovine serum albumin (BSA) and the ΔUCS mutant of UNC-45, deficient in myosin binding, as negative controls. When co-incubated at 27 °C, aberrant myosin did not influence the running behavior of BSA and ΔUCS during SEC, as both control proteins eluted at the same retention volume as in the absence of myosin. In conclusion, the SEC experiments indicate that unfolded myosin binds to UNC-45 in a specific manner (Fig. 4b). We also analyzed the identified UNC-45 N801A mutant, deficient in myosin folding, in the in vitro assay (Supplementary Fig. 3b). Unexpectedly, the N801A mutant still interacts with damaged myosin under the applied conditions. These data suggest that the UNC-45 mutant protein, which has a partially obstructed myosin-binding site, can still bind to substrate, however, does not fully support the folding of myosin in the cell (Fig. 2b, d and Supplementary Fig. 2a).

The established in vitro interaction assay furthermore allowed us to address the molecular basis underlying the UNC-45 *ts*-mutations, which are among the most studied motility defects in *C. elegans*[10,11,13]. Of note, our cellular chaperone assay revealed that the UNC-45 *ts*-mutations have a negative effect on the myosin folding activity, however, did not resolve whether the *ts*-mutations directly impair the chaperone activity or exert rather indirect effects, destabilizing for example the UNC-45 protein. With regards to the latter point, we carried out circular dichroism (CD) measurements to determine the melting temperature ($T_m$) and thus stability of the purified *ts*-proteins (Fig. 4c). The CD data demonstrate that all four UNC-45 *ts*-proteins exhibit the same or even slightly higher thermal stability than the wt chaperone. Moreover, the E781K mutant displayed a slightly different unfolding profile pointing to potential structural differences in the folded chaperone (Supplementary Fig. S6a). It should be also noted that it was even possible to crystallize

various *ts*-mutants at room temperature (19 °C) and determine their structure, as described later.

When addressing the interaction with myosin in the SEC-based assay, we observed that all UNC-45 *ts*-mutants showed a reduced ability to form a complex with the unfolded myosin, as judged by the smaller protein amounts co-eluting with the myosin substrate (Fig. 4c). To further confirm that the impaired chaperone activity of the *ts*-mutants is not due to an unstable UNC-45 protein, we repeated the in vitro chaperone assay, this time monitoring the interaction of *ts*-mutants with pre-misfolded myosin at low temperature. After pre-incubating myosin at 27 °C, the heat-damaged protein was mixed with different UNC-45 proteins at 4 °C and the samples were analyzed by SEC as described above. For the wt chaperone, we observed a shift of the UNC-45 fraction towards the HMW peak that should contain the chaperone-substrate complex (Supplementary Fig. S6b). In contrast, none of the *ts*-proteins shifted into the HMW fraction further confirming that the *ts*-mutants, present in a folded state, cannot interact with damaged myosin (Supplementary Fig. S6c). Together, the cellular folding assay, the CD data and the SEC-based reconstitution analysis demonstrate that the *ts*-mutations L559S, G427E, E781K and L822F have a direct effect on the myosin-binding function of UNC-45, abrogating the interaction with the heat-damaged myosin substrate. Thus, our data provide compelling evidence that the temperature-induced motility defect of *ts*-worms is caused by the impaired chaperone activity of UNC-45 rather than being the consequence of an unstable or misfolded *ts*-protein.

**Structural features of UNC-45 *ts*- and deletion mutants**. To address the structural basis of the impaired chaperone function of UNC-45 *ts*-mutants L822F and G427E, and UNC-45 ΔTPR, we analyzed the crystal structures of the three mutant proteins (Supplementary Table 2). The two UNC-45 *ts*-mutants L822F and G427E crystallized in the form of a linear filament, having the same periodicity (85 Å), hydrophobic interface and overall domain organization as the previously characterized wt oligomer. However, when we compared the UCS domains of wt and *ts* UNC-45, we noted two major differences. First, the *ts* UCS domains adopt different conformations, as most clearly seen for the C-terminal portion of the UCS domain that moves about 10–15 Å (Fig. 5a, b). Notably, a previous structural analysis revealed two different UCS domain conformations of the wt protein, in which the armadillo (ARM) repeats 14–17 compose either a curved (PDB code: 5mzu) or an extended (PDB code: 4i2z) UCS fold (Fig. 5a) and ARM repeat 13 serves as molecular hinge within the domain. Likewise, the relative orientation of the N- and C-terminal UCS subdomains varies markedly between G427E and wt UNC-45 (Fig. 5b). Importantly, this rearrangement remodels the myosin-binding canyon, thus providing different electrostatic and spatial constraints for recognizing and accommodating potential substrate proteins. Moreover, as revealed by the L822F structure, the entire UCS domain is able to move en-bloc towards the TPR/central domain backbone of the UNC-45 molecule (Fig. 5c). Such gross domain rearrangement is expected to alter the geometry and thus the functionality of UNC-45 complexes with partner chaperones, in addition to modulating substrate binding and release via the UCS domain.

A second difference of wt UNC-45 and *ts* mutants comprises the dynamics of the UCS domain. For the wt protein, the two crystal forms capturing the alternate UCS conformations were grown from the same crystallization drops, suggesting that the observed states occur in equilibrium in solution[19,24]. In contrast, the two *ts* mutants crystallized exclusively in one state, even when testing a large number of crystals for an alternate crystal form. For G427E, the introduced mutation, which lies in the linker

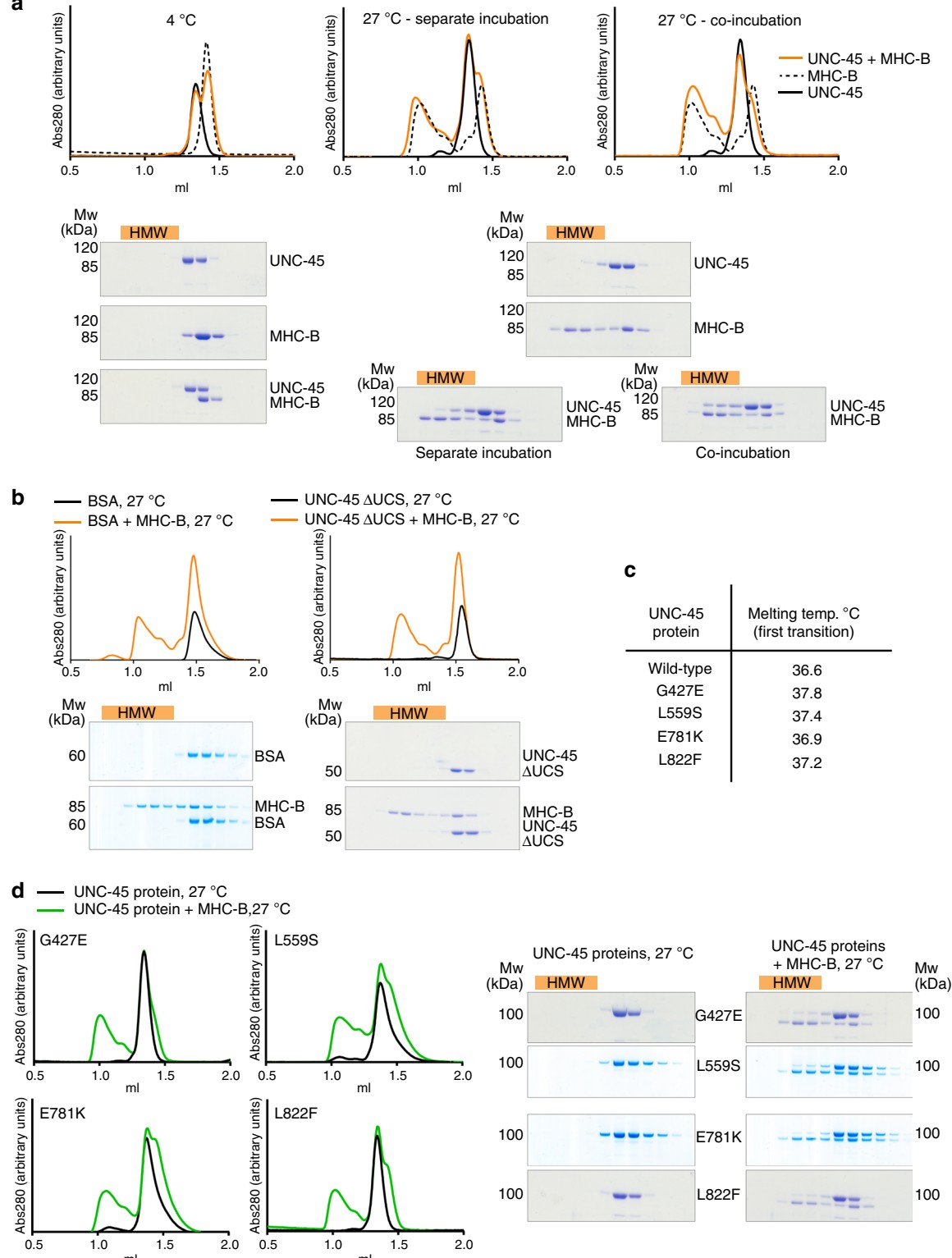

**Fig. 4** Interaction of UNC-45 proteins and myosin in vitro. **a** Analytical SEC and SDS-PAGE analysis of UNC-45 and myosin incubated together at 4 °C or at 27 °C and incubated separately at 27 °C and mixed before the SEC run. Misfolded myosin and bound UNC-45 proteins elute earlier from the SEC column, indicated as high molecular weight fractions (HMW, orange bar). **b** Controls for interaction studies shown in **a**. BSA or UNC-45 ΔUCS were incubated with myosin at 27 °C. **c** Melting temperature as determined by CD spectroscopy for the indicated UNC-45 proteins. **d** Same analysis as in **a** for the indicated UNC-45 *ts*-mutants and myosin, incubated together at 27 °C

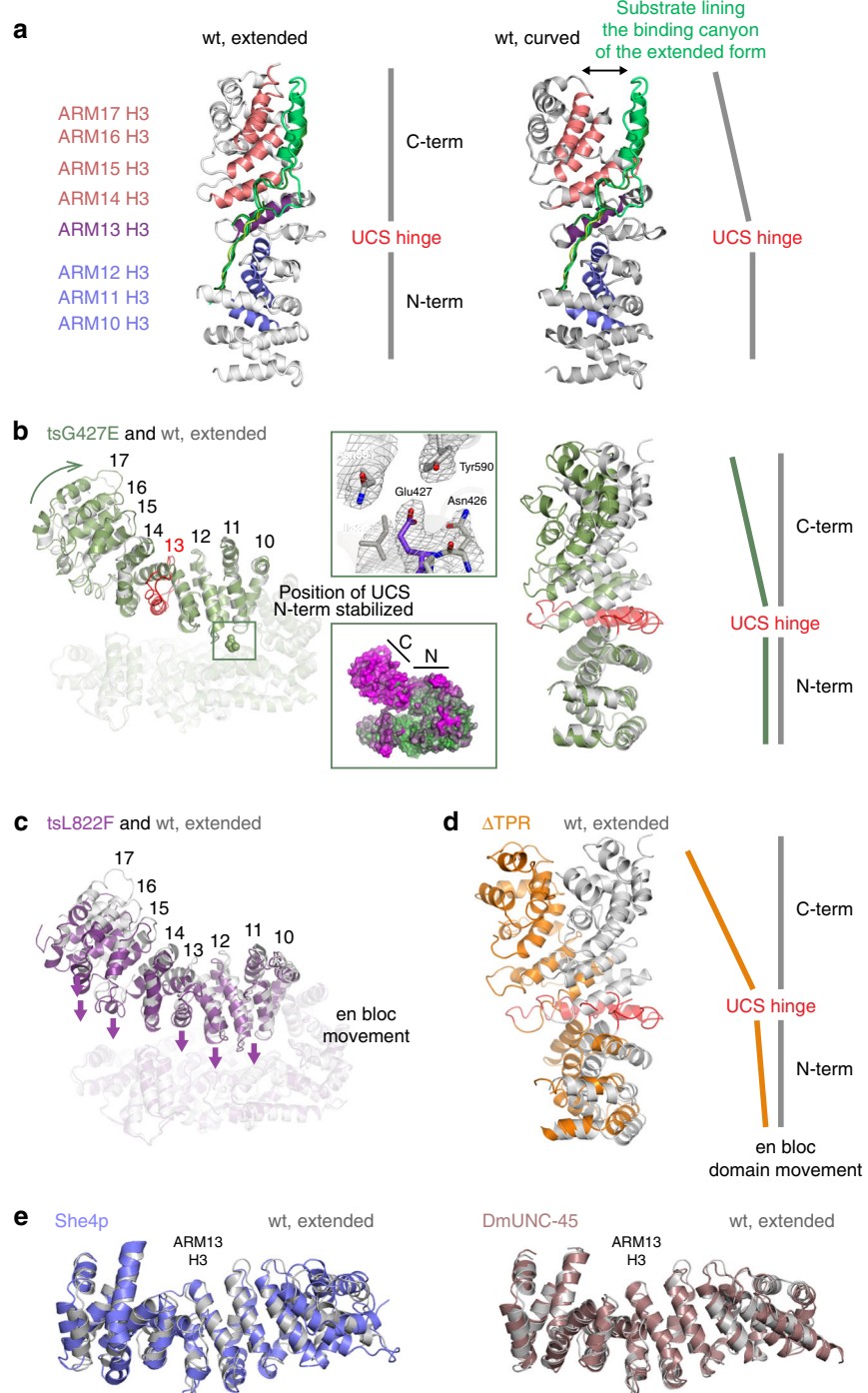

**Fig. 5** Structural analysis of UNC-45 mutant proteins. **a** Structural alignment of two different wild-type UNC-45 structures—PDB code: 4i2z, extended UCS domain conformation, PDB code 5mzu, curved UCS conformation. The three helices of each ARM repeat are organized such that helices H3 (colored) form the bottom of the myosin-binding canyon, whereas helices H1 and H2 compose the border of the molecular cleft. Substrates binding to the myosin binding-canyon are illustrated by overlay of ligands co-crystallized with the ARM domain of beta-catenin. **b** Structural alignment of wild-type UNC-45 (PDB code: 4i2z) and UNC-45 tsG427E—side view (left panel), top view (right panel). The top inset shows the E427 residue overlaid with 2fofc density contoured at 1σ. The bottom inset shows the thermal motion (b) factors displayed on the surface of the tsG427E mutant. N- and C-terminal subdomains of the UCS domain are indicated (green – rigid, magenta – flexible). **c** Structural alignment of wild-type UNC-45 (PDB code: 4i2z) and tsL822F. Arrows indicate the en bloc movement of the entire UCS domain towards the TPR/central domain. **d** Structural alignment of wild-type UNC-45 (PDB code: 4i2z) and UNC-45 ΔTPR molecule B. All structural alignments were performed with residues 1–523 as reference. **e** Structural alignment of wild-type UNC-45 (PDB code: 4i2z) and the UCS proteins She4p (PDB code: 3opb) and DmUNC-45 (PDB code: 3now). The center of the UCS myosin binding canyon (ARM 13 H3) is highlighted

region connecting the central and the UCS domain, induces small, local changes that propagate over the entire UCS domain (Fig. 5b). Movement of ARM repeats 11–15 toward the central domain results in the remodeling of the interface between UCS and central domain (Fig. 5b). This compaction of the UNC-45 molecule leads to additional van-der-Waals contacts between the domains, which in concert rigidify the UCS fold as suggested by its reduced dynamics (Fig. 5b). Likewise, the L822F *ts* mutation causes local changes inducing movement of the UCS domain towards the TPR/central backbone (Fig. 5c). As for the G427E mutant, the established inter-domain contacts reduce the conformational flexibility of the UCS domain, which seems to be a common property of the analyzed *ts* mutants.

Finally, we determined the crystal structure of the UNC-45 ΔTPR construct that represents a *C. elegans* UNC-45 variant in monomeric, non-filamentous form. The asymmetric unit of the ΔTPR crystal form is built up by two molecules. In molecule A, the UCS domain adopts an extended conformation similar to wt UNC-45, whereas molecule B contains a UCS domain that is strongly bent, with the entire domain and even more strongly its C-terminal half (ARM repeats 14–17) extending away from the central domain (Fig. 5d). These data further highlight the pronounced structural flexibility of the myosin-binding UCS domain, which is held in place by non-covalent interactions with the adjacent central and TPR domains. Accordingly, deleting the TPR domain releases the flexible UCS domain to adopt various conformations, two of which were captured in the present crystal form.

To further characterize the identified UCS conformations, we performed a DALI search[33] and looked for related protein structures in the PDB database. This analysis revealed distinct sets of structural homologs (Supplementary Table 3). In notable contrast to UNC-45, the identified β-catenin, importin α, and karyopherin α seem to occur only in one major conformation. As the identified ARM proteins use distinct geometries to recognize and bind their cognate substrate proteins, the observed UCS shapes of UNC-45 could, in analogy, differ in their myosin-binding properties. To evaluate the possible predominance of one state, we compared the UCS domain of UNC-45 with that of the structurally and functionally related Drosophila Unc45 and yeast She4 proteins[25,37]. Remarkably, the UCS domains of DmUnc45 and She4p fit well to the extended fold of wt UNC-45, ΔTPR molecule A, and L822F, as also reflected in the Z-scores of the DALI search (Fig. 5e and Supplementary Table 3). Since *C. elegans* UNC-45, DmUnc45 and She4p were crystallized in different crystal forms, the corresponding UCS conformation should not reflect a crystal packing artifact. We thus presume that the extended UCS conformation represents an important functional state of the myosin-binding site that is preferentially attained by the native chaperone. Taken together, our analysis demonstrates that the UCS domain can adopt distinct conformations leading to the rearrangement of its N- and C-terminal halves and thus of the engaged myosin-binding canyon. Moreover, we show that the myosin-binding deficient mutations, though being located at different portions of the UNC-45 molecule, influence the interaction of the UCS domain with the central and TPR domains, thereby modulating the overall conformation as well as flexibility of the myosin-binding module.

## Discussion

Although the role of UNC-45 in myosin assembly has been extensively studied, insights into the molecular mechanism of the chaperone is limited[29]. To better understand its myosin folding function, we established an orthogonal protein folding assay in insect cells, characterizing the interplay of the *C. elegans* muscle

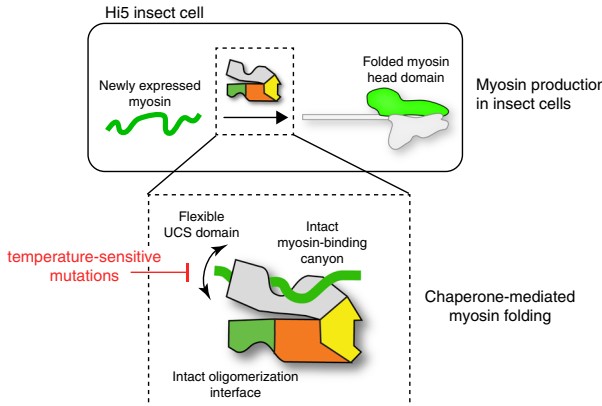

**Fig. 6** UNC-45 chaperone function in folding the myosin head domain. Fully functional myosin can be obtained by co-expression with its cognate chaperone UNC-45 in insect cells. UNC-45's ability to support myosin folding strongly depends on the flexibility of its UCS domain, the functionality of the myosin-binding canyon and its oligomerization properties. UNC-45 *ts-mutations* limit this conformational flexibility and therefore interfere with myosin binding and chaperone activity

myosin MHC-B with its cognate chaperone UNC-45 (Fig. 6). Moreover, the obtained muscle myosin II enabled us to directly monitor complex formation with UNC-45 in vitro using a SEC-based assay and to address the underlying effect of the long-studied UNC-45 *ts*-mutations, extensively characterized in vivo[10]. In fact, it is the limited ability of the *ts*-mutants to bind unfolded myosin (Fig. 4d) that should underlie their *unc* phenotype. Given the success in elucidating the long-sought molecular basis of the UNC-45 *ts*-mutations, we expect that the described in vitro and in vivo chaperone assays will be valuable tools for studying myosin folding in general. For example, the assays could be applied to characterize additional factors involved in myosin maturation, such as Smyd1b[38–40] or the Hsp70/Hsp90 co-factors[14,19,28,41–44].

The present findings are also of interest for the recombinant production of muscle myosin. As the biochemical and structural characterization of myosin II is critical for understanding muscle function in health and disease, its recombinant production has been in the focus of research for decades. However, any attempts to express the motor domain of striated muscle myosin II in bacterial or insect cells failed[27,45–47]. So far, insect cell expression systems have been used to produce non-muscle myosin[48], whereas co-expression of UNC-45 has further enabled production of an unconventional myosin[49]. Recombinant striated muscle myosin II could only be produced in C2C12 myoblasts[21,50,51] or specific Drosophila lines[52]; however, both systems are not suitable for large-scale protein expression and require specific expertise in working with the expression host. The UNC-45/myosin expression system presented here enables the production of large amounts of the motor domain of striated muscle myosin II, MHC-B, which may overcome this limitation. Importantly, protein expression and purification rely on simple standard procedures that can be carried out in short time and easily scaled up to obtain the required amounts of myosin II for detailed biochemical and structural studies, as exemplified by the presented crystal structure of MHC-B, which is one of the most highly resolved myosin structures in the Protein Data Bank. Given the similarity of human cardiac myosin II and MHC-B, our system offers an efficient way in producing myosin mutants that may reveal molecular details underlying human myopathies[53–55].

Chaperones ensure the proper folding of newly synthesized proteins and prevent the formation of toxic protein aggregates

during stress situations. Here, we show that the myosin specific chaperone UNC-45 is essential for both processes. Structural data of various functional mutants clarify the role of the distinct UNC-45 domains in this process providing important mechanistic insight into its myosin folding activity.

The N-terminal TPR domain of UNC-45 is a well-characterized protein–protein interaction domain that serves as the major binding site for Hsp70 and Hsp90[14,19,56]. The developed cellular and in vitro chaperone assays revealed that the Hsp70/Hsp90 binding mutant UNC-45 K82E[19] did not have any negative effect on the yield of functional myosin suggesting that a direct interaction between the UNC-45 TPR domain and its partner chaperones is not required for the folding of the myosin II motor domain. Contrary, a UNC-45 mutant lacking the entire TPR domain did not support myosin folding in insect cells. These data indicate that the TPR domain of UNC-45 is not only critical to coordinate the interplay with partner chaperones, but has additional functions: (1) In the UNC-45 oligomer, the TPR domain is essential to connect neighboring molecules[19]. As shown in this study, the L121W mutation, which prevents oligomer formation, negatively affects the amount of produced myosin. (2) Our structural data suggest that the TPR domain obtains an additional intra-molecular scaffolding role. It undergoes van-der-Waals contacts with the UCS domain, thereby arranging the myosin binding site to accommodate substrate. Accordingly, formation of UNC-45 oligomers via the TPR domain may be critical to set up a functional UCS myosin-binding site, thus linking UNC-45 oligomerization to its myosin folding activity.

The most common characteristic for molecular chaperones is a structurally flexible binding site for client proteins[57,58,59]. The present data suggest this concept holds also true for the myosin-binding domain of UNC-45. The UCS domain adopts an extended ARM repeat scaffold that is penetrated by a long canyon serving as protein binding site. Our structural data reveal the capability of the UCS domain to adopt distinct defined conformations. Together, the wt, G427E, L822F, and ΔTPR crystal structures depict four different UCS conformations highlighting the pronounced dynamicity of the myosin-binding domain (Fig. 5). Even though the functionality of the characterized conformations remains to be elucidated, the switch between them should be carefully controlled as several states appear to be non-functional preventing the binding of the myosin substrate. In conclusion, the biochemical and structural data suggest that the conformational flexibility of the UCS domain is an important determinant of substrate selectivity. It appears that the plastic UCS domain offers a further layer of regulation. Induced-fit substrate binding, allosteric effects of the TPR domain, higher-order oligomer formation as well as post-translational modifications could thus have a strong impact on the function of the UNC-45 chaperone in bringing muscle proteins in shape.

## Methods
**Insect cell protein expression and purification**. MHC-B (1–790) and NMY-2 (1–796) were cloned from *C. elegans* cDNA into the pH promotor site of the pFastBac DUAL vector (Invitrogen) with a C-terminal His₆-tag. FLAG-Hsp70 (*C. elegans* HSP-1), FLAG-Hsp90 (*C. elegans* DAF-21) or UNC-45-Strep were added into the p10 promotor site of the pFastBac DUAL vector. Previously described constructs served as DNA templates[19]. All primers used are listed in Supplementary Table 4. Baculovirus was produced by transfection of SF9 cells (Expression Systems) with FuGENE HD (Promega). For expression, High Five cells (Expression Systems) at a density of $1 \times 10^6$ cells ml⁻¹ were infected with baculovirus (P1 amplification) and incubated for 72 h at 21 °C. The cells were collected by centrifugation, lysed by freeze-thaw, and resuspended in 50 mM Tris, pH 7.5, 300 mM NaCl (buffer A), containing 20 mM imidazole. The cleared cell lysate was loaded onto a NiNTA column (GE Healthcare) equilibrated with buffer A. After washing the column, myosin and associated UNC-45 were eluted with buffer A, containing 150 mM imidazole. The elution was applied to a Superdex 200 column equilibrated

with 20 mM Tris pH 8.0, 150 mM NaCl. For crystallization purposes, this buffer was optimized to 20 mM Tris pH 8.5, 20 mM NaCl.

For small scale myosin purifications (shown in Fig. 2), a pellet from 50 ml co-expression culture was lysed in 4 ml of 50 mM Tris pH 7.5, 150 mM NaCl, 20 mM imidazole containing benzonase. The cleared cell lysate was applied to 100 μl OPUS RoboColumns (Repligen) packed with Ni Sepharose 6 Fast Flow (GE Healthcare) using a Janus Liquid handling system (Perkin Elmer). The resin was washed with 2 ml of 50 mM Tris pH 7.5, 150 mM NaCl, 20 mM imidazole and 2 ml of 50 mM Tris pH 7.5, 150 mM NaCl, 40 mM imidazole and finally eluted in 400 μl of 50 mM Tris pH 7.5, 150 mM NaCl, 150 mM imidazole. Fifty microliter of the elution fraction was following loaded onto a Superdex 200 5/150 GL column pre-equilibrated with 50 mM Tris pH 7.5, 150 mM NaCl.

**Bacterial protein expression and purification**. UNC-45 point mutants (G427E, L559S, E781K, L822F, N801A) were generated by site-directed mutagenesis of the wild-type protein in pET21a. All primers used are listed in Supplementary Table 4. UNC-45 G427E SeMet protein was produced in B834-DE3 cells (Novagen) as previously described[19]. All native UNC-45 proteins were expressed and purified as previously described[19,24]. Briefly, expression was performed in BL21(RIL) cells (Stratagene) induced with IPTG. The three-step purification procedure included NiNTA and anion exchange chromatography and a final SEC step in 20 mM Tris pH 8.0, 150 mM NaCl.

**Actin preparation**. Rabbit actin was purified by the method of Lehrer and Kewar[60] with minor modifications. Briefly, no ATP was added at the final step to polymerize actin and residual amounts of nucleotide were removed by sequential centrifugation of actin and dialysis steps. Actin was labeled with pyrene (pyr-actin) as previously described[61].

**CD spectroscopy**. CD experiments were essentially performed as previously described[24]. Briefly, spectra (190–260 nm) of UNC-45 proteins at 0.7 mg ml⁻¹ were recorded on a Chirascan plus CD spectrometer (Applied Photophysics). Data were analyzed with the Global3 evaluation software.

**Western blot analysis of insect cell lysates**. For comparing the effect of different UNC-45 mutants on the solubility of MHC-B, UNC-45 proteins were cloned from previously described constructs[19,24] and bacterial expression constructs described above into a pIDC derivative encoding an N-terminal Twin-Strep-tag (former One-Strep-tag) followed by a 3C protease cleavage site. The MHC-B motor domain (1–790) was cloned into pACEBac1 with a C-terminal fusion to a His₆-tag. FLAG-CeHsp90 and FLAG-CeHsp70 were cloned into pACEBac1 derivatives. All primers used are listed in Supplementary Table 4. The MultiBac was generated by Golden Gate assembly or Cre recombination (Geneva Biotech) and bacmids were generated by transposition of *E. coli* DH10EMBacY (Geneva Biotech). Baculovirus production and protein expression was performed as described above. For expressions at 18 °C, cells were harvested after 96 h. High Five cells were harvested, lysed by freeze-thaw and resuspended in 50 mM sodium phosphate buffer pH 8, 300 mM NaCl. Whole cell lysates and cleared cell lysates (supernatant after centrifugation at 21130 g for 30 min at 4 °C) were analyzed by Western blot using anti-FLAG (Sigma F1804, used 1:1,000), anti-Penta-His (Qiagen, used 1:1,000) and anti-CeUNC-45 antibodies (used 1:10,000)[24]. For Supplementary Fig. 2a, samples were normalized to soluble UNC-45 levels.

**Pull-down of *C. elegans* UNC-45 from insect cells**. High Five cells (Expression Systems) at a density of $1 \times 10^6$ cells ml⁻¹ were infected with baculovirus (P1 amplification) and incubated for 72 h at 21 °C. A pellet from 7.5 ml expression culture was lysed in 2 ml of 50 mM Tris pH 7.5, 20 mM NaCl containing benzonase. Cleared cell lysate was applied to 100 μl OPUS RoboColumns (Repligen) packed with Strep-Tactin Superflow (IBA Lifesciences) using a Janus Liquid handling system (Perkin Elmer). The resin was washed with 3 ml of 50 mM Tris pH 7.5, 20 mM NaCl and UNC-45 was eluted with 100 μl of 50 mM Tris pH 7.5, 20 mM NaCl, 5 mM d-Desthiobiotin. Elutions were analyzed by western blot using anti-Hsp70 (Santa Cruz Biotechnology, SC59572, used 1:1000), anti-Hsp90 (Abcam [EPR16621-67] (ab203126), used 1:250) and anti-Strep (Qiagen 34850, used 1:2500) antibodies.

**Analytical size exclusion**. UNC-45 proteins or BSA and myosin were incubated at a final concentration of 5 μM at 4 °C or 27 °C for 60 min. For the SEC analysis in Supplementary Fig. 6, myosin was incubated at 27 °C for 60 min followed by incubation with UNC-45 (wild-type or *ts*-mutant protein) on ice for 60 min. After incubation samples were applied to a S200 PC 3.2 column in 20 mM Tris pH 8.0, 150 mM NaCl. Indicated fractions were analyzed by SDS-PAGE.

**F-actin co-sedimentation assay**. F-actin was mixed with myosin (plus and minus 30 mM ATP) to a final concentration of 10 and 2.5 μM, respectively. The samples were centrifuged at 150,000 g, for 60 min at 4 °C. Supernatant and pellet fractions were analyzed by SDS-PAGE.

**Stopped-flow measurements**. Stopped-flow experiments to measure MHC-B single turnover were performed with a TgK stopped-flow apparatus (excitation wavelength 280 nm, emission filtered at 320 nm). The experimental data were fitted to a single exponential using Kinetic Studio, and the observed rate analyzed with Prism (GraphPad).

**Transient kinetics**. Fast kinetic data was performed as previously described[62,63]. All measurements were performed at 20 °C in 20 mM MOPS, 25 mM KCl, 5 mM $MgCl_2$, 1 mM $NaN_3$ at pH 7.0, unless indicated otherwise. Rapid-mixing experiments were completed in triplicate using a High-Tec Scientific SF-61 DX2 stopped-flow system. Transient kinetic traces were initially fitted with Kinetic Studio (TgK Scientific) and subsequently plotted with Origin (OriginPro). For the single turn-over experiment, 10 μM MHC-B and 5 μM ATP stock solutions (concentration before mixing) were prepared in 20 mM MOPS pH 7, 25 mM KCl, 5 mM $MgCl_2$, 1 mM $NaN_3$. For the actin affinity experiment, the amplitude dependence of the MHC-B concentration was fitted to the standard quadratic equation for a binding isotherm[64].

$$\alpha = \frac{[M] + K_D + [A]_0 - \sqrt{\left([M] + K_D + [A]_0\right)^2 - \frac{4}{[M][A]_0}}}{2[A]_0} \quad (1)$$

Where $\alpha$ = fraction of actin bound to MHC-B, $[M]$ = concentration of MHC-B, $K_D$ = the dissociation constant of MHC-B for actin and $[A]_0$ = concentration of actin. For experiments probing the actin-myosin interaction with ATP and ADP the fluorescence signal for pyrene-labelled actin was recorded, which has an excitation wavelength at 365-nm and the emission was detected after passing through KV389-nm cut-off filter. In the absence of actin, intrinsic tryptophan fluorescence was monitored by excitation at 295-nm and emission detected with a WG-320 cut-off filter.

**ATPase assay and actin-stimulated ATPase activity**. ATPase activity was determined by a coupled enzymatic reaction[65]. In all, 2 μM MHC-B was incubated with 37.5 U $ml^{-1}$ pyruvate kinase, 42.9 U $ml^{-1}$ lactate dehydrogenase, 0.25 mM NADH, 15 mM phosphoenolpyruvate, and 2 mM ATP in a buffer containing 20 mM MOPS pH 7.0, 100 mM KCl, 5 mM $MgCl_2$. The absorption at 340 nm was recorded for 30 min using a Pherastar FS plate reader. The experiment was performed in the presence of increasing concentrations of actin (ranging from 0 to 90 μM) and corrected by subtracting the ATPase activity of actin. The Michaelis-Menten equation was fit to the data to determine the maximal activity ($V_{max}$) and the association constant of actin for myosin ($K_m$) using Prism (GraphPad). For the activity measurement of MHC-B produced with different UNC-45 mutants, the experiment was performed with 90 μM actin concentration.

**Crystallization and data collection**. Crystals of MHC-B were grown in 96-well plates set up with a Mosquito robot (TTP Labtech) in 0.1 M MES/imidazole pH 6.5, 10% w/v PEG 4000, 20% v/v glycerol, 0.02 M Morpheus alcohol mix (0.2 M 1,6-hexanediol, 0.2 M 1-butanol, 0.2 M (RS)-1,2-propanediol, 0.2 M 2-propanol, 0.2 M 1,4-butanediol, 0.2 M 1,3-propanediol). Crystals of the UNC-45 G427E mutant were obtained by manually mixing 2 μl protein (200 μM) with 0.3 μl additive (1.0 M $(NH_4)_2SO_4$) and 1 μl reservoir solution (0.1 M HEPES pH 8.0, 5% Tacsimate, 15% PEG MME 5000). Crystals of UNC-45 L822F and UNC-45 ΔTPR were grown in 96-well plates set up with a Mosquito robot (TTP Labtech) in 0.1 M MES/NaOH pH 6.5, 0.2 M ammonium acetate, 30% glycerol ethoxylate and 0.1 M Tris pH 8.5, 32% glycerol ethoxylate, respectively. All crystals were grown by sitting drop vapor diffusion: MHC-B crystals at 4 °C, UNC-45 crystals at 19 °C. MHC-B, UNC-45 L822F and ΔTPR were directly frozen in liquid nitrogen, whereas UNC-45 G427E crystals were shortly incubated in reservoir solution containing 25% ethylene glycol, mounted into 90°-bent loops and frozen in liquid nitrogen as previously described[19]. Diffraction data were collected at the European Synchrotron Radiation Facility (ESRF) (wavelength: 0.979 Å): UNC-45 G427E anomalous diffraction dataset at beamline ID 29, UNC-45 L822F dataset at beamline ID 14-4; UNC-45 ΔTPR dataset at ID 23-1. Diffraction data for MHC-B were collected at Deutsches Elektronen-Synchrotron (DESY) (wavelength: 0.979 Å), beamline P13. Data were integrated with XDS and scaled with SCALA or XSCALE[66,67].

**Structure solution and refinement**. The crystal structure of MHC-B was solved by molecular replacement with Phaser using chicken smooth muscle myosin motor domain as search model (PDB code: 5m05)[68]. The structure was built with Coot and O, and the final refinement performed using PHENIX[69–71]. The crystal structure of UNC-45 G427E was solved by using experimental SAD phases. Refinement proceeded smoothly in cycles of model rebuilding with O and crystallographic refinement with CNS[71,72]. The final model was refined with PHENIX[69]. The UNC-45 L822F and ΔTPR structures were solved by molecular replacement (MR) with PHASER[68]. The wild-type UNC-45 (PDB code: 4i2z) structure was split into three parts comprising residues 5–507, 525–760, and 761–930 to create an optimal search model for the MR of the UNC-45 L822F structure. For solution of the UNC-45 ΔTPR structure, the previous UNC-45 structure (PDB code: 4i2z) was divided into two fragments 135–507 and 525–930. Molecule A was clearly defined, while residues 719–828 and 829–930 had to be

placed as separate units into the density obtained for molecule B. Initial structural refinement of the UNC-45 L822F and ΔTPR structures was performed with CNS[72]. Following rebuilding of the structure using COOT and O, the final refinement was performed using PHENIX[69–71]. Structure figures were generated with PYMOL[73]. Data collection and refinement statistics are shown in Supplementary Table 2. Ramachandran statistics for the determined structures are as follows: UNC-45 L822F - favored 96.9%, allowed 3.1%, outliers 0.0%; UNC-45 G427E - favored 94.4%, allowed 5.3%, outliers 0.3%; UNC-45 ΔTPR - favored 97.3%, allowed 2.7%, outliers 0.0%; MHC-B - favored 97.7%, allowed 2.2%, outliers 0.1%.

**Reporting summary**. A reporting summary for this Article is available as a Supplementary Information file. Further information on research design is available in the Nature Research Reporting Summary linked to this article.

## Data availability

Data supporting the findings of this papermanuscript are available from the corresponding authors upon reasonable request. The source data underlying Figs. 1b, c, e, 2b–d, 3a–d, and 4a, b, d and Supplementary Figs. 1–3 and 6b, c are provided as a Source Data file. Coordinates and structure factors were deposited in the Protein Data Bank under accession codes 6QDL (UNC-45 L822F), 6QDK (UNC-45 G427E), 6QDM (UNC-45 ΔTPR), and 6QDJ (MHC-B).

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

## Acknowledgements

We thank the VBCF Protech Facility for the support with biophysical measurements. R.A. was supported by the Austrian Science Fund (FWF) DK W1258-B22. C.J. and M.A.G. were supported by NIH grant GM029090. The IMP is funded by Boehringer Ingelheim.

## Author contributions

D.H., A.L. and N.F. performed the UNC-45 biochemistry. A.L, N.F. and R.K. performed the insect cell studies. N.F., R.A., C.J. and M.A.G. performed the myosin biochemistry and kinetic analysis. D.H., R.A., A.M., L.D., L.G. and T.C. performed the structural analysis. D.H. and T.C. outlined the study and prepared the paper with input from all authors.

## Competing interests

The authors declare no competing interests.
