## [Peer Review File · Nature Communications]

Reviewers' Comments:

Reviewer #1:

Remarks to the Author:

This is an interesting manuscript describing the expression of a *C. elegans* skeletal muscle myosin 2 (MHC-B) in insect cells when co-expressed with the *C. elegans* chaperone UNC-45. It is the first successful expression of a striated muscle myosin in this system, albeit unfortunately not from a mammal, which would have been a greater accomplishment. Importantly, the system they have pioneered allowed them to show the direct effect of the ts-mutations on the myosin folding activity of UNC45 that was previously unknown. In my opinion, the most important use of this system is to further probe the role of UNC45 and other co-chaperones in myosin folding; the applicability to making human disease mutations in this myosin is limited given the poor sequence conservation with mammalian striated muscle and the known differing functional effects of mutant phenotypes on the exact myosin sequence (e.g. alpha versus beta cardiac myosin).

Fig.1 The expression of Hsp70 and 90 appears to be much more robust in the control experiment than in the unknown. Why? In panel c, how can the soluble fraction be more than what was observed in the whole cell lysate (UNC45 and Hsp90)? Presumably equal amounts were loaded, or maybe the lanes are mis-labeled. Please explain. Also, it would be informative to compare co-expression with UNC-45 alone to that of a co-expression with Unc45, Hsp90, and Hsp70 to see if the myosin isolated from the two scenarios has identical properties.

Fig. 2 and Fig 3 are mislabeled in the figures but referred to correctly in text. Fig. 2 is Fig. 3 and vice versa.

Fig. 2 (TPR mutants) Direct evidence that baculovirus expressed K82E UNC45 cannot bind Hsp90/70 should be added to support their conclusion that direct binding is not required. Some of the results (e.g. N801A) look quite different between expression 1 and 2, although it is true that it is less than WT in both cases.

The observation that UNC45 oligomerization is also needed for expression of soluble fragments weakens the argument in their previous 2013 Cell paper regarding the coupling of myosin head spacing in a filament with this oligomerization.

Fig. 3 (biochemical characterization) The biochemical characterization is inadequate and the weakest part of the paper that needs to be remedied prior to publication. Based on these data, I would never have predicted that the protein would crystallize. Particularly because expressed myosin yields are not limiting there is no reason to not have more complete data. The gel figure shows incomplete release from actin in the presence of ATP. The actin-activated ATPase activity needs to be performed at higher actin concentrations to obtain K_m and V_{max} values, standard parameters obtained from this assay. The single turnover basal rate of 0.073/s does not agree with the high value of $\sim 1/s$ seen at zero actin in the steady-state assay. Why?

Fig. 3 Although the beta-cardiac was the closest structural homolog of the *C. elegans* myosin, its 59% sequence identity precludes any useful application of this system to understand mutations in cardiac myosin implicated in disease.

Reviewer #2:

Remarks to the Author:

Summary

In *C. elegans*, the folding and assembly of the striated muscle myosin (UNC-54) relies on the specific chaperone UNC-45. The structure of this chaperone has been extensively studied by this group as well as the mechanism of its interaction with the co-chaperones Hsp90 and Hsp70. This

earlier work included the identification of a novel organization of UNC-45 in tandem oligomers that support thick filament formation and stability in *C. elegans*. This paper describes a reconstituted insect cell expression system to evaluate the mechanism of action of UNC-45 on the folding of the MHC-B (UNC-54) motor domain by co-expression. Using this system they have completed a mutational analysis of UNC-45 and determined crystal structures of temperature sensitive mutations of UNC-45 to advance the understanding of the folding mechanism. The results focus on the flexible myosin-binding UCS domain, and they claim to have uncovered the molecular basis of temperature sensitive UNC-45 mutations. Unfortunately, the data fall significantly short of this goal.

Critique

To monitor the interaction between *C. elegans* UNC-45 and MHC-B they have developed a system for co-expression of chaperones and MHC-B using Baculovirus vectors and insect cells (High Five cells). Expression of many of the UNC-45 mutations in *C. elegans* leads to embryonic lethality complicating mechanistic studies; this is not a problem with this insect cell system. They could also trim the myosin down to just the essential motor domain, residues 1-790 in this case, producing a truncated monomeric enzyme. They show that in the absence of UNC-45 the MHC-B motor domain (myB-MD) was expressed but not soluble; however, co-expression with wild type UNC-45 produced a soluble protein that was purified, subsequently crystallized, and crystal structure solved. Folding of the myB-MD was not supported by co-expression with Hsp90 or Hsp70 alone, two known UNC-45 co-chaperones. Only UNC-45 promoted the maturation of the myB-MD presumably complemented by the endogenous insect cell Hsp90/Hsp70 chaperones. This point should have been demonstrated but was not. The folding assay that they developed involved co-expression of myB-MD with a variety of UNC-45 truncations and mutations followed by determining if the myB-MD was soluble after lysis of the insect cells. Soluble protein was judged as successful folding. This criteria for folding is too limited and needs biochemical support. Is the soluble protein folded or in a complex with the co-chaperones UNC-45, Hsp90, and Hsp70? How do you define folded vs. soluble in this system?

They attempted to demonstrate that the purified myB-MD isolated from this system is a biochemically active motor and the best evidence of that is the crystallization and structure solution. However, they include a biochemical characterization in Figure 3, which they have mislabeled Figure 2, and those data are unconvincing. The basal myosin ATPase activity is measured as 0.07 s⁻¹ however, the actin activation of the myB-MD ATPase is minimal (~40% above basal) over the range of 0 – 5 μM F-actin. The ATPase of most muscle myosin MD constructs is activated by 5-20 fold or greater over this range of actin concentrations. In addition, they didn't do the activation curve over a sufficient range to establish a V_{max} for myosin or KATPase for actin. Also, there is no indication that they have corrected the measured ATPase activity for the F-actin ATPase. The actin ATPase activity alone could account for the linear rise that they report. Finally, the myB-MD does not show good ATP sensitive binding to actin. The actin pellet in the presence of ATP contains about 50% of the myB-MD, suggesting a large fraction of ATP insensitive binding known in the field as 'dead-heads'.

They emphasize how effective co-expression with UNC-45 is for production of myB-MD and suggest that this is an invaluable tool for studying myosin mutations. They seem to have missed the fact that this approach has been used successfully with baculovirus in insect cells before to co-express vertebrate unc45b with mouse myosin-15 and isolate and characterize the unconventional myosin motor domain. The biochemistry of the expressed recombinant motor protein in that paper demonstrates how to assay ATPases, do in vitro motility, and do single molecule mechanics and should have been cited and used as a guide here (Bird et al., 2014. Chaperone-enhanced purification of unconventional myosin 15, a molecular motor specialized for stereocilia protein trafficking. *Proc Natl Acad Sci* 111:12390-12395). Furthermore, despite the structural similarity of the myB-MD to other myosin MD's it is in the end a *C. elegans* protein domain evolved for a role in that organism and not a model of human cardiac myosin, no matter how much protein you can make.

The biggest problem with the data has to do with the analysis of the UNC-45 ts mutations. A key discovery they report is that they have uncovered the molecular basis of temperature sensitive UNC-45 mutations. However, they fail to consider the 'temperature sensitivity' of the mutations. They should have stated the permissive temperature range (normally 15-20°C in *C. elegans*) and the restrictive temperature (usually above 25°C) and addressed how the "ts mutations" behave under these conditions. E.g. the insect cell expression was done at 21°C, at the borderline for thermal sensitivity in *C. elegans*. Some of the mutations failed at this temp and other produced some soluble protein. Was that a result of differing stability of the ts mutations and would they have performed better at a lower temperature (e.g. 15-18°C). Why didn't they evaluate the optimal temperature for the mutations? In addition, the in vitro assays with the "ts proteins" were done at 27°C, above the restrictive temperature, but this is not mentioned or addressed. If they fail in *C. elegans* at the restrictive temperature (e.g. 25°C), why wouldn't they expect failure in vitro at 27°C? They note that the "ts mutations" were stable at 19°C because they crystallized the proteins at this temperature, but this is within the permissive temperature range and doesn't address the thermal stability of the proteins. Analysis of thermal stability using e.g. CD or calorimetry would address the stability issue.

As a result I don't think they have addressed the mechanism of "temperature sensitivity" at all. That leaves the structural analysis of the UNC-45 ts mutants and the various conformations that they have observed without a foundation for evaluation and, the extrapolation of the structural flexibility to mechanism is just speculation.

We would like to thank the reviewers for their comments and have addressed their concerns point-by-point below.

Reviewer #1 (Remarks to the Author):

This is an interesting manuscript describing the expression of a *C. elegans* skeletal muscle myosin 2 (MHC-B) in insect cells when co-expressed with the *C. elegans* chaperone UNC-45. It is the first successful expression of a striated muscle myosin in this system, albeit unfortunately not from a mammal, which would have been a greater accomplishment. Importantly, the system they have pioneered allowed them to show the direct effect of the ts-mutations on the myosin folding activity of UNC45 that was previously unknown. In my opinion, the most important use of this system is to further probe the role of UNC45 and other co-chaperones in myosin folding; the applicability to making human disease mutations in this myosin is limited given the poor sequence conservation with mammalian striated muscle and the known differing functional effects of mutant phenotypes on the exact myosin sequence (e.g. alpha versus beta cardiac myosin).

1.1) Fig.1 The expression of Hsp70 and 90 appears to be much more robust in the control experiment than in the unknown. Why?

The two anti-FLAG blots visualizing expression of CeHsp70/90 are not directly comparable, as they were performed separately. The difference in signal intensity arises from technical differences in performing the two western blots. Still, according to these data, the two proteins are co-expressed at substantial amounts in the indicated experiments.

1.2) In panel c, how can the soluble fraction be more than what was observed in the whole cell lysate (UNC45 and Hsp90)? Presumably equal amounts were loaded, or maybe the lanes are mis-labeled. Please explain.

In panel c, only a third of the whole cell lysate with respect to the soluble fraction was applied (i.e. not equal to the amount of the soluble fraction, to avoid over-exposure of bands when developing the western blots). A corresponding note is now included in the figure legend of **Fig. 1c**.

1.3) Also, it would be informative to compare co-expression with UNC-45 alone to that of a co-expression with Unc45, Hsp90, and Hsp70 to see if the myosin isolated from the two scenarios has identical properties.

To address the potential influence of the nematode partner chaperones Hsp70 and Hsp90 on producing functional MHC-B, we followed the advice of this referee and compared the biochemical properties of MHC-B resulting from co-expressing UNC-45 alone and UNC-45 together with Hsp70/Hsp90. As illustrated in **Fig. 1e**, the SEC profiles of the two MHC-B samples nicely overlap pointing to similar amounts and quality of the produced myosin. Moreover, we compared the actin-inducible ATPase activity and could show that the two MHC-B samples exhibit virtually identical enzymatic activities. The new findings are described on page 6:

*“Finally, when purifying MHC-B by NiNTA and size exclusion chromatography (SEC) and testing its actin-induced ATPase activity, we did not observe major differences in the amount and functionality of the myosin motor co-expressed with UNC-45 alone or together with *C. elegans* Hsp70 and Hsp90 (Fig. 1e and Supplementary Fig. 2).”*

2.1) Fig. 2 and Fig 3 are mislabeled in the figures but referred to correctly in text. Fig. 2 is Fig. 3 and vice versa.

The labelling of the figures has been corrected.

2.2) Fig. 2 (TPR mutants) Direct evidence that baculovirus expressed K82E UNC45 cannot bind Hsp90/70 should be added to support their conclusion that direct binding is not required.

To address this point, we performed pull-down studies with *C. elegans* UNC-45 expressed in insect cells and monitored the appearance of endogenous Hsp70 and Hsp90 in the eluted fractions. As shown in **Fig.1d**, the new data clearly demonstrate that UNC-45 and Hsp70/Hsp90 interact with each other and that this interaction is specific. When introducing the point mutation K82E into the TPR domain of UNC-45, the co-elution of the partner chaperones is strongly impaired. Moreover, in vivo and in vitro data indicate that the K82E mutant exhibit full chaperone function thus supporting our original hypothesis, as described on page 6:

“The data further imply that the nematode UNC-45 can team up with the insect cell chaperone machinery required for myosin folding, given that additional co-expression of the cognate partner chaperones Hsp70 and Hsp90 from C. elegans was not required to obtain soluble myosin (Fig. 1c). Indeed, we could pull-down endogenous Hsp70 and Hsp90 together with C. elegans UNC-45 from insect cell lysates (Fig. 1d). This interaction is abolished upon deletion of the UNC-45 TPR domain or mutating a key residue (K82E) in the Hsp70/90 binding groove, while deletion of the UCS domain does not impact the interaction with the partner chaperones (Fig. 1d). Finally, when purifying MHC-B by NiNTA and size exclusion chromatography (SEC) and testing its actin-induced ATPase activity, we did not observe major differences in the amount and functionality of the myosin motor co-expressed with UNC-45 alone or together with C. elegans Hsp70 and Hsp90 (Fig. 1e and Supplementary Fig. 2).”

2.3) Some of the results (e.g. N801A) look quite different between expression 1 and 2, although it is true that it is less than WT in both cases.

Minor differences are observed between individual co-expression experiments (which was the reason why we presented both Western blots in the original manuscript), but - as noted by the reviewer - the reduced chaperone activity of the UNC-45 mutant is apparent in both repeats. To corroborate these results, we conducted a third co-expression/Western-blot experiment, which clearly confirmed the tendencies of the first two repeats. The data are now shown in the new **Supplementary Fig. 3a**. To further demonstrate the impact of the introduced mutations on the UNC-45 chaperone capacity, we purified MHC-B from all co-expression trials and subjected the samples to a SEC analysis, monitoring specifically the amount of folded protein (new **Fig. 2d**). The in vitro characterization of the purified MHC-B confirmed tendencies seen in the cellular chaperone assay and indicated that modifying the substrate-binding site of the UCS domain (e.g. N801A mutation) impairs the production of folded myosin.

2.4) The observation that UNC45 oligomerization is also needed for expression of soluble fragments weakens the argument in their previous 2013 Cell paper regarding the coupling of myosin head spacing in a filament with this oligomerization.

This is a very valid point that - thanks to the referees' comments - could be clarified during the revision process. In the original manuscript, we highlighted the effect of the L121W mutant in yielding less soluble MHC-B than wild-type UNC-45. However, when characterizing the obtained MHC-B fraction further, we realized that the NiNTA-purified fraction contains a major fraction of folded, functional myosin, as seen in the qualitative SEC analysis (new **Fig. 2d**) and in ATPase assays (new **Supplementary Fig. 2**). In summary, the new data indicate that the introduced chaperone mutations should be judged by two criteria - that is (1) the total amount of **soluble** protein produced and (2) the amount of **folded** myosin in the soluble fraction (please see also our detailed comments at 4.2). According to these criteria, the L121W had a negative impact on the overall yield of soluble myosin, but apparently is still capable to promote myosin folding. We are thus very thankful for the two referees for their insightful comments for better characterizing the developed myosin expression system. To account for the new findings, we have modified the parts describing the effect of the L121W mutation, which still acts as a myosin chaperone, fitting to the 2013 proposed scaffolding function of UNC-45 chains.

*“To test the importance of oligomer formation for myosin folding, we analyzed UNC-45 L121W, a single-site mutant known to disrupt the interface stabilizing UNC-45 chains. As seen in the insect cell chaperone assay, the L121W mutation led to a decrease in the amount of soluble MHC-B head domain (**Supplementary Fig. 3a**). However, in our SEC analysis, a significant amount of the soluble MHC-B elutes as folded protein (**Fig. 2d**), suggesting that UNC-45 oligomerization is not required for the maturation of the myosin motor domain.” (Page 9/10)*

3.1) Fig. 3 (biochemical characterization) The biochemical characterization is inadequate and the weakest part of the paper that needs to be remedied prior to publication. Based on these data, I would never have predicted that the protein would crystallize. Particularly because expressed myosin yields are not limiting there is no reason to not have more complete data.

Although a detailed functional analysis of MHC-B was not a major focus of our study, we now performed a comprehensive biochemical characterization of the purified myosin motor. We included, for example, a detailed kinetic characterization of MHC-B, in comparison to human beta cardiac myosin that served as our reference system. In summary, our new data shown in **Fig. 3**, **Supplementary Fig. 5** and **Supplementary Table 1** clearly demonstrate that our expression system allows to produce functional myosin, which exhibits similar kinetic properties to other muscle myosins.

3.2) The gel figure shows incomplete release from actin in the presence of ATP.

To provide a more appropriate picture representing MHC-B functionality, we optimized the conditions of the co-sedimentation assay, in particular by increasing the ATP concentration as this variable has an effect on the ATP/ADP ratio due to MHC-B catalyzed ATP hydrolysis. Under the new conditions, we observed complete release of myosin from F-actin.

3.3) The actin-activated ATPase activity needs to be performed at higher actin concentrations to obtain K_m and V_{max} values, standard parameters obtained from this assay.

We agree and thus repeated the mentioned ATPase assays accordingly. For this purpose, we purified native rabbit actin and analyzed its stimulatory effect on myosin ATPase activity, using a broader range of actin concentrations. As shown in the new **Fig. 3c**, containing 6 replicates (assays were conducted for 2 distinct MHC-B purification 3 times), we measured a V_{max} of 0.257 molecules of

ATP per myosin head per second, and an average K_m of 23.1 μM . According to our new data, the basal ATPase activity is stimulated about 6 times at high actin concentrations, a number that agrees to other muscle myosins (e.g. Miller et al, 2007, JMB).

Given the protein quality of the MHC-B myosin motor, its abundance and its ideal properties for conducting stopped-flow measurements (strong tryptophan signal as well as being an efficient quencher of pyrene-actin fluorescence), we performed a detailed characterization of its kinetic parameters using a stopped-flow setup. The new experimental data are summarized in **Fig. 3d**, **Supplementary Fig. 5** and **Supplementary Table 1** and include a comparison with human beta cardiac myosin.

3.4) The single turnover basal rate of 0.073/s does not agree with the high value of $\sim 1/\text{s}$ seen at zero actin in the steady-state assay. Why?

Regarding the raised concern about the discrepancy of the basal ATPase rates of myosin observed in the single-turnover and steady-state assays, we would like to note that the steady-state values were normalized to the myosin activity in the absence of actin, which was set to "1". To make the two assays better comparable, we thus decided to include absolute rate numbers for the steady-state experiment that comprises 6 replicates (3 technical repeats done with MHC-Bs purified from 2 different insect cell expressions). The resulting basal rate number of 0.043/s is in the same range as the single-turnover number (0.073/s), with the minor difference being expected for the distinct experimental setups.

3.5) Fig. 3 Although the beta-cardiac was the closet structural homolog of the *C. elegans* myosin, its 59% sequence identity precludes any useful application of this system to understand mutations in cardiac myosin implicated in disease.

While agreeing with the reviewer that the overall sequence identity is only 59%, the conservation in the active site is much higher (approximately 80-90%), and the protein structures superimpose very well (RMSD of 2.1 Å for 696 Ca atoms). We would also like to mention that a *Drosophila melanogaster* model of cardiomyopathy has been recently reported by the Bernstein lab for the study of the disease relevant K146N mutant (Kronert et al., Prolonged cross-bridge binding triggers muscle dysfunction in a *Drosophila* model of myosin-based hypertrophic cardiomyopathy, 2018, eLife). Since *D. melanogaster* identity with the human homologue is with 61% comparable to the *C. elegans* MHC-B, and given that *C. elegans* is a good genetic model for muscle biology, we believe that the developed system will have a similar impact on myosin research as the *Drosophila* model system. Having said this, we agree with the argument raised by both referees and have toned down our original claim. Still, we would like to highlight the potential of MHC-B as a model protein for addressing the function and assembly of muscle myosin, as stated on page 13:

"In contrast to the human protein, which has to be produced in a special mouse cell line requiring time- and resource-demanding procedures, the biochemical and structural analysis of MHC-B was performed with a recombinant protein that can be produced in almost unlimited amounts in insect cells (15 mg myosin per liter cell culture) and is easily accessible to genetic manipulations. Considering the unprecedented yields and the overall similarity to mammalian muscle myosins, including the human beta-cardiac variant, the recombinant MHC-B should represent a valuable model system to address the assembly, function and regulation of muscle myosins in basic and in medical research."

Reviewer #2

Summary

In *C. elegans*, the folding and assembly of the striated muscle myosin (UNC-54) relies on the specific chaperone UNC-45. The structure of this chaperone has been extensively studied by this group as well as the mechanism of its interaction with the co-chaperones Hsp90 and Hsp70. This earlier work included the identification of a novel organization of UNC-45 in tandem oligomers that support thick filament formation and stability in *C. elegans*. This paper describes a reconstituted insect cell expression system to evaluate the mechanism of action of UNC-45 on the folding of the MHC-B (UNC-54) motor domain by co-expression. Using this system they have completed a mutational analysis of UNC-45 and determined crystal structures of temperature sensitive mutations of UNC-45 to advance the understanding of the folding mechanism. The results focus on the flexible myosin-binding UCS domain, and they claim to have uncovered the molecular basis of temperature sensitive UNC-45 mutations. Unfortunately, the data fall significantly short of this goal.

Critique

To monitor the interaction between *C. elegans* UNC-45 and MHC-B they have developed a system for co-expression of chaperones and MHC-B using Baculovirus vectors and insect cells (High Five cells). Expression of many of the UNC-45 mutations in *C. elegans* leads to embryonic lethality complicating mechanistic studies; this is not a problem with this insect cell system. They could also trim the myosin down to just the essential motor domain, residues 1 -790 in this case, producing a truncated monomeric enzyme. They show that in the absence of UNC-45 the MHC-B motor domain (myB-MD) was expressed but not soluble; however, co-expression with wild type UNC-45 produced a soluble protein that was purified, subsequently crystallized, and crystal structure solved. Folding of the myB-MD was not supported by co-expression with Hsp90 or Hsp70 alone, two known UNC-45 co-chaperones.

4.1. Only UNC-45 promoted the maturation of the myB-MD presumably complemented by the endogenous insect cell Hsp90/Hsp70 chaperones. This point should have been demonstrated but was not.

To address the collaboration of UNC-45 with Hsp70 and Hsp90 from insect cells, we performed pull-down studies. These experiments clearly show an interaction between *C. elegans* UNC-45 and insect cell Hsp70 and Hsp90 proteins. The interaction we observed in these experiments is specific since removal of the UNC-45 TPR domain and introduction of the K82E mutation abolish the interaction of UNC-45 with its insect cell partner chaperones. The new findings are presented in the new **Fig. 1de** and on page 6

*“The data further imply that the nematode UNC-45 can team up with the insect cell chaperone machinery required for myosin folding, given that additional co-expression of the cognate partner chaperones Hsp70 and Hsp90 from *C. elegans* was not required to obtain soluble myosin (Fig. 1c). Indeed, we could pull-down endogenous Hsp70 and Hsp90 together with *C. elegans* UNC-45 from insect cell lysates (Fig. 1d). This interaction is abolished upon deletion of the UNC-45 TPR domain or mutating a key residue (K82E) in the Hsp70/90 binding groove, while deletion of the UCS domain does not impact the interaction with the partner chaperones (Fig. 1d). Finally, when purifying MHC-B by NiNTA and size exclusion chromatography (SEC) and testing its actin-induced ATPase activity, we*

did not observe major differences in the amount and functionality of the myosin motor co-expressed with UNC-45 alone or together with C. elegans Hsp70 and Hsp90 (Fig. 1e and Supplementary Fig. 2)."

4.2. The folding assay that they developed involved co-expression of myB-MD with a variety of UNC-45 truncations and mutations followed by determining if the myB-MD was soluble after lysis of the insect cells. Soluble protein was judged as successful folding. This criteria for folding is too limited and needs biochemical support. Is the soluble protein folded or in a complex with the co-chaperones UNC-45, Hsp90, and Hsp70? How do you define folded vs. soluble in this system?

We agree to this point of criticism and developed an improved method to characterize the folding status of the produced MHC-B myosins. For this purpose, we applied a 3-step purification strategy involving (1) cell lysis and removal of cell remnants by centrifugation, (2) NiNTA pull-down to enrich His-tagged myosins and (3) an analytical SEC gel filtration run for separating monomeric, folded from aggregated and misfolded myosin molecules. With regards to the used nomenclature, we now refer to **soluble** protein present in the supernatant after cell lysis and removal of aggregates by centrifugation. Upon NiNTA and analytical SEC, we could quantify the amounts of **folded** myosin monomers. Finally, we tested the activity of the folded MHC-Bs in an ATPase activity assay, in the absence and presence of actin, to determine the **functional** state. The latter assay revealed that all soluble and folded myosins, which are present in the monomeric SEC peak, exhibit a comparable, actin-inducible ATPase activity (**Supplementary Fig. 2**) and should thus represent functional myosins. The detailed analysis of all co-expressions done during the revision (solubility analysis, purification by NiNTA/SEC and ATPase assays) are summarized in **Fig. 2**, **Supplementary Fig. 2** and **Supplementary Fig. 3**.

With regards to our original classification we noticed two differences: First, for the Y750W mutant, where myosin was efficiently produced in insect cells, we only observed a minor fraction in the "folded" SEC peak, suggesting that the chaperone activity of this mutant is reduced. Second, an opposite scenario holds true for the L121W mutant, where we observed only little amounts of soluble protein in the cell. However, this portion is strongly enriched in folded myosin molecules, highlighting the chaperone activity of the mutant. In conclusion, thanks to the remarks of this referee, we were able to improve our cellular chaperone assay by adding an in vitro characterization of the produced myosin. The adapted text in the manuscript reads now as follows:

"While Y750W supported the production of soluble myosin in the cell (Supplementary Fig. 3a), the yield of functional MHC-B that could be purified is smaller than for wild-type UNC-45 (Fig. 2d). These data suggest that even subtle structural changes in proximity to the UCS myosin-binding site affect UNC-45 chaperone function." (Page 8)

"To test the importance of oligomer formation for myosin folding, we analyzed UNC-45 L121W, a single-site mutant known to disrupt the interface stabilizing UNC-45 chains. As seen in the insect cell chaperone assay, the L121W mutation led to a decrease in the amount of soluble MHC-B head domain (Supplementary Fig. 3a). However, in our SEC analysis, a significant amount of the soluble MHC-B elutes as folded protein (Fig. 2d), suggesting that UNC-45 oligomerization is not required for the maturation of the myosin motor domain." (Page 9)

4.3. They attempted to demonstrate that the purified myB-MD isolated from this system is a biochemically active motor and the best evidence of that is the crystallization and structure solution.

However, they include a biochemical characterization in Figure 3, which they have miss-labeled Figure 2 (corrected), and those data are unconvincing. The basal myosin ATPase activity is measured as 0.07 s⁻¹ however, the actin activation of the myB-MD ATPase is minimal (~40% above basal) over the range of 0 – 5 μM F-actin. The ATPase of most muscle myosin MD constructs is activated by 5-20 fold or greater over this range of actin concentrations. In addition, they didn't do the activation curve over a sufficient range to establish a V_{max} for myosin or KATPase for actin.

The same points of concern were raised by the first referee. Please have a look at our answers there (3.1.-3.5.). In brief, we have now conducted a comprehensive analysis of the MHC-B motor, characterizing its interaction with actin in a detailed stopped-flow analysis. Also, the improved steady-state analysis, which was conducted following the advice of both referees, fits now well to the single-turnover data and reveals a 6-fold stimulation of ATPase activity in the presence of F-actin. We consider the introduced data as comprehensive and convincing, and matching nicely the *in vivo* chaperone assay as well as the structural data.

4.4. Also, there is no indication that they have corrected the measured ATPase activity for the F-actin ATPase. The actin ATPase activity alone could account for the linear rise that they report.

The measurements had been corrected for the F-actin ATPase activity (measured by an actin-only control), however, this was not properly mentioned in the methods section. We have now added a corresponding statement:

“The experiment was performed in the presence of increasing concentrations of actin (ranging from 0 to 90 μM) and corrected by subtracting the ATPase activity of actin.” (Page 29)

4.5. Finally, the myB-MD does not show good ATP sensitive binding to actin. The actin pellet in the presence of ATP contains about 50% of the myB-MD, suggesting a large fraction of ATP insensitive binding known in the field as 'dead-heads'.

We fine-tuned our experimental conditions, in particular the ATP concentration during co-sedimentation experiments, as this variable has an effect on the ATP/ADP ratio due to MHC-B catalyzed ATP hydrolysis, allowing us to present the functional status of the produced MHC-B more properly (we now see a complete release). Still, we would like to note that this experiment was originally included to demonstrate that we had active protein, and not the level of activity. Moreover, the latter property can be more accurately assessed from the pyrene signals, which show the expected degree of quenching on forming a 1:1 complex and complete recovery of signal on addition of ATP (new **Fig. 3d**).

4.6. They emphasize how effective co-expression with UNC-45 is for production of myB-MD and suggest that this is an invaluable tool for studying myosin mutations. They seem to have missed the fact that this approach has been used successfully with baculovirus in insect cells before to co-express vertebrate unc45b with mouse myosin-15 and isolate and characterize the unconventional myosin motor domain. The biochemistry of the expressed recombinant motor protein in that paper demonstrates how to assay ATPases, do *in vitro* motility, and do single molecule mechanics and should have been cited and used as a guide here (Bird et al., 2014. Chaperone-enhanced purification of unconventional myosin 15, a molecular motor specialized for stereocilia protein trafficking. Proc Natl Acad Sci 111:12390-12395).

We apologize for having missed this citation. We now specifically refer to the work of Bird et al on Page 21:

“Despite this strong interest, any attempts to express the motor domain of muscle myosin II in bacterial or insect cells failed^{27, 48, 49, 50}. So far, insect cell expression systems have been used to produce non-muscle myosin⁵¹, whereas co-expression of UNC-45 has further enabled production of an unconventional myosin⁵².”

4.7. Furthermore, despite the structural similarity of the myB-MD to other myosin MD's it is in the end a *C. elegans* protein domain evolved for a role in that organism and not a model of human cardiac myosin, no matter how much protein you can make.

The same point was raised by the other referee. As discussed under 3.5, we would like to refer to a recent work from the Bernstein lab, in which they established a *Drosophila melanogaster* model of cardiomyopathy, studying the K146N mutant (Kronert et al., Prolonged cross-bridge binding triggers muscle dysfunction in a *Drosophila* model of myosin-based hypertrophic cardiomyopathy, 2018, eLife). Since *D. melanogaster* identity with the human homologue is comparable to the *C. elegans* MHC-B (61% vs 59% identical, respectively), and because *C. elegans* is a good genetic model for muscle biology, we believe that the introduced system will have a similar impact on myosin research as the referred to *Drosophila* system. Having said this, we agree to the argument raised by both referees and toned-down our original claim. Still, we would like to highlight the potential of MHC-B as a model protein for addressing the function and assembly of muscle myosin, as stated on page 13:

“In contrast to the human protein, which has to be produced in a special mouse cell line requiring time- and resource-demanding procedures, the biochemical and structural analysis of MHC-B was performed with a recombinant protein that can be produced in almost unlimited amounts in insect cells (15 mg myosin per liter cell culture) and is easily accessible to genetic manipulations. Considering the unprecedented yields and the overall similarity to mammalian muscle myosins, including the human beta-cardiac variant, the recombinant MHC-B should represent a valuable model system to address the assembly, function and regulation of muscle myosins in basic and in medical research.”

4.8. The biggest problem with the data has to do with the analysis of the UNC-45 ts mutations. A key discovery they report is that they have uncovered the molecular basis of temperature sensitive UNC-45 mutations. However, they fail to consider the ‘temperature sensitivity’ of the mutations. They should have stated the permissive temperature range (normally 15-20°C in *C. elegans*) and the restrictive temperature (usually above 25°C) and addressed how the “ts mutations” behave under these conditions. E.g. the insect cell expression was done at 21°C, at the borderline for thermal sensitivity in *C. elegans*. Some of the mutations failed at this temp and other produced some soluble protein. Was that a result of differing stability of the ts mutations and would they have performed better at a lower temperature (e.g. 15-18°C). Why didn't they evaluate the optimal temperature for the mutations?

In the revised manuscript we specifically mention the permissive (15-18C) and restrictive (25C) temperatures, at which experiments with ts-worms were previously conducted (page 10). Despite this temperature dependence observed for in vivo experiments with *C. elegans*, our new data unequivocally demonstrate that the ts-mutations do not affect the stability of the UNC-45 protein itself but its chaperone function:

(1) First, we carried out insect cell co-expression studies at 18°C, which lies in the permissive temperature range with respect to the UNC-45 ts-worms and still supported survival of the High Five

insect cells. Also, at this lower temperature, the ts-mutants did not improve the solubility of myosin, as shown in **Supplementary Fig. 3b**. Clearly, we did not observe a temperature-dependent effect in making soluble, folded myosin.

In addition, the in vitro assays with the “ts proteins” were done at 27°C, above the restrictive temperature, but this is not mentioned or addressed. If they fail in *C. elegans* at the restrictive temperature (e.g. 25°C), why wouldn't they expect failure in vitro at 27°C?

(2) Second, to further rule out any destabilizing effects of incubating the ts-mutants at 27°C, we performed the SEC interaction studies with heat-damaged myosin at 4°C, a temperature where the ts mutants are stable over days. These data are presented in **Supplementary Fig. 8bc** and demonstrate that all ts-mutants, present in a folded state, do not interact with misfolded myosin.

They note that the “ts mutations” were stable at 19°C because they crystallized the proteins at this temperature, but this is within the permissive temperature range and doesn't address the thermal stability of the proteins. Analysis of thermal stability using e.g. CD or calorimetry would address the stability issue.

(3) Third, and most importantly, we carried out CD measurements to determine the melting temperature (T_m , reflecting protein stability) of the 4 ts-mutant proteins. The obtained T_m values (shown in **Fig. 4c**) clearly demonstrate that the ts-mutants are as stable as the wild-type UNC-45 protein. We therefore did not expect “failure” of the ts-mutants at 27°C.

As a result I don't think they have addressed the mechanism of “temperature sensitivity” at all. That leaves the structural analysis of the UNC-45 ts mutants and the various conformations that they have observed without a foundation for evaluation and, the extrapolation of the structural flexibility to mechanism is just speculation.

Taken together, the new data provide compelling evidence that the UNC-45 ts-mutants are stably folded proteins. Strikingly, even when conducting experiments at the permissive temperature range (e.g. co-expression at 18°C and analytical SEC interaction studies at 4°C), the ts-mutations abolish the UNC-45 chaperone activity. We thus conclude that the introduced single-point mutations do not affect protein stability, but represent functional mutations, most likely impairing the interaction with the unfolded myosin substrate. These new findings are described in the following paragraphs:

*“Temperature-sensitive (ts)-mutants: In *C. elegans*, four distinct UNC-45 point mutations (G427E, L559S, E781K, L822F) have been identified that underlie the so-called uncoordinated (unc) ts-phenotype^{10, 11, 13, 32}. Ts-worms grown at the permissive temperature of 15-18°C during their larval development show no defects in myosin thick filament formation, while ts-worms grown at the restrictive temperature of 25°C exhibit motility defects and a distorted sarcomere organization^{10, 11}. Though known for decades, the mechanistic details underlying the ts-phenotype of UNC-45 have remained elusive. Strikingly, in our cellular chaperone assay, all G427E, L559S, E781K and L822F ts-mutations strongly reduced the production of soluble myosin (**Fig. 2b**). Applying the minimal amounts of soluble protein to a SEC column did not recover any folded protein (**Fig. 2d**), indicating that the ts-mutants cannot promote maturation of the native myosin motor. To test whether the defect in chaperone function relates to a reduced stability of the ts-proteins themselves, we repeated the cellular folding assays at a lower temperature. Our standard expressions were performed at 21°C, which we further reduced to 18°C, a low temperature that still supported survival of insect cells and that corresponds to the permissive temperature in growing ts-worms. Also, at 18°C we observed the*

same solubility defect for myosin co-expressed with the *ts*-mutants (**Supplementary Fig. 3b**). These results suggest that the UNC-45 *ts*-mutants, which were expressed at the same level as the wt protein, affect the chaperone function of UNC-45 rather than its stability.” (Page 10)

“Additionally, we carried out circular dichroism (CD) measurements to determine the melting temperature (T_m) and thus stability of the purified *ts*-proteins (**Fig. 4c**). The CD data demonstrate that all four UNC-45 *ts*-proteins exhibit the same or even slightly higher thermal stability than the wt chaperone. Moreover, the E781K mutant displayed a slightly different unfolding profile pointing to potential structural differences in the folded chaperone (**Supplementary Fig. S8a**). It should be also noted that it was even possible to crystallize various *ts*-mutants at room temperature (19°C) and determine their structure, as described later. When addressing the interaction with myosin in the SEC-based assay, we observed that all UNC-45 *ts*-mutants showed a reduced ability to form a complex with the unfolded myosin, as judged by the smaller protein amounts co-eluting with the myosin substrate (**Fig. 4c**). To further confirm that the impaired chaperone activity of the *ts*-mutants is not due to an unstable UNC-45 protein, we repeated the *in vitro* chaperone assay, this time monitoring the interaction of *ts*-mutants with pre-misfolded myosin at low temperature. After pre-incubating myosin at 27°C, the heat-damaged protein was mixed with different UNC-45 proteins at 4°C and the samples were analyzed by SEC as described above. For the wt chaperone, we observed a shift of the UNC-45 fraction towards the HMW peak that should contain the chaperone-substrate complex (**Supplementary Fig. S8b**). In contrast, none of the *ts*-proteins shifted into the HMW fraction further confirming that the *ts*-mutants, present in a folded state, cannot interact with damaged myosin (**Supplementary Fig. S8c**). Together, the cellular folding assay, the CD data and the SEC-based reconstitution analysis demonstrate that the *ts*-mutations L559S, G427E, E781K and L822F have a direct effect on the myosin-binding function of UNC-45, abrogating the interaction with the heat-damaged myosin substrate. Thus, our data provide compelling evidence that the temperature-induced motility defect of *ts*-worms is caused by the impaired chaperone activity of UNC-45 rather than being the consequence of an unstable or misfolded *ts*-protein.” (Page 15)

Reviewers' Comments:

Reviewer #1:

Remarks to the Author:

The authors have made a serious effort to improve the paper with the inclusion of more data. Most of my concerns have been addressed. I still believe this work will be most useful to probe chaperone function rather than myosin function. A few remaining points for consideration:

1. Figs 1b and 1 c are mixed up although referred to correctly in text and legend.
2. One thing I still object to is the idea that this is a good model system for human disease, although their statements have been tempered in the revised version. One rationale is that "the nucleotide binding site of MHC-B is virtually identical to that of human beta-cardiac myosin" (lines 289-291). Yes, but this is in contrast to their functional data (Supplementary Fig.1) that shows an order of magnitude difference in the ADP affinity for both myosin and actomyosin in *C. elegans* versus human beta-cardiac myosin, showing that the two myosins differ significantly. One can't have it both ways.
3. p.13 line 296 Delete "unprecedented". The yields are higher than C2C12 cells but not that unusual for many proteins expressed in insect cells. I also would not describe 15 mg protein as "almost unlimited" amounts.
4. Line 334 change to more scientific wording than "...dragging it towards the HMW species."
5. Other "muscle" myosins such as smooth muscle myosin II have also been successfully expressed in insect cells using endogenous folding machinery, not just non-muscle myosins. Throughout the text, they should clarify muscle myosin as "striated" muscle myosin.
6. p.4 define UFD-2

Reviewer #2:

Remarks to the Author:

The revised manuscript has addressed to the satisfaction of this reviewer all of the major concerns raised in the initial review. This is a substantially improved presentation and of broad scientific interest. The incorporation of the revised text addressing the concerns and additional data in the paper and supplement markedly improves the manuscript.

There is one labeling issue that needs to be addressed. Line 102 refers the NMY-2 folding in Fig. 1b, but in the Figure panel c is labeled (I believe correctly) as NMY-2. Accordingly line 114 refers to Fig. 1c for MHC-B and that is panel b in the Figure 1. The legend for the Figure 1 (line 756) has these two panels incorrectly identified. This confusion needs attention.

We appreciate the positive response of the reviewers and have addressed their final concerns, as detailed in our point-by-point response below.

Reviewer #1 (Remarks to the Author):

The authors have made a serious effort to improve the paper with the inclusion of more data. Most of my concerns have been addressed. I still believe this work will be most useful to probe chaperone function rather than myosin function. A few remaining points for consideration:

1. Figs 1b and 1 c are mixed up although referred to correctly in text and legend.

The figure panels were exchanged to fit to the text and legend.

2. One thing I still object to is the idea that this is a good model system for human disease, although their statements have been tempered in the revised version. One rationale is that “the nucleotide binding site of MHC-B is virtually identical to that of human beta-cardiac myosin” (lines 289-291). Yes, but this is in contrast to their functional data (Supplementary Fig.1) that shows an order of magnitude difference in the ADP affinity for both myosin and actomyosin in *C. elegans* versus human beta-cardiac myosin, showing that the two myosins differ significantly. One can't have it both ways.

Given the high similarity of human beta cardiac myosin and MHC-B, we still propose that the *C. elegans* MHC-B, for which an efficient expression and purification system is presented in this study, will be a valuable model to study muscle myosin function in basic and medical research. Indeed we observe a difference in ADP affinity, however, this should not preclude MHC-B from serving as an important research tool in studying myosin-related human disease. We have rephrased our statement to more cautiously describe the suitability of the model system: “Given the similarity of human cardiac myosin II and MHC-B, our system offers an efficient way in producing myosin mutants that may reveal molecular details underlying human myopathies.”

3. p.13 line 296 Delete “unprecedented”. The yields are higher than C2C12 cells but not that unusual for many proteins expressed in insect cells. I also would not describe 15 mg protein as “almost unlimited” amounts.

The word “unprecedented” on p13 line 296” and other direct references to novelty were removed. The yield of 15 mg per liter culture is now described as “large amounts”.

4. Line 334 change to more scientific wording than “...dragging it towards the HMW species.”
As part of shortening the manuscript, this statement was deleted.

5. Other “muscle” myosins such as smooth muscle myosin II have also been successfully expressed in insect cells using endogenous folding machinery, not just non-muscle myosins. Throughout the text, they should clarify muscle myosin as “striated” muscle myosin. To clarify this point we specifically refer to the MHC-B myosin used in this study as ‘striated muscle myosin’ in the introduction and in the discussion section where expression and purification of different types of myosins are described.

6. p.4 define UFD-2

The abbreviation was defined.

Reviewer #2 (Remarks to the Author):

The revised manuscript has addressed to the satisfaction of this reviewer all of the major concerns raised in the initial review. This is a substantially improved presentation and of broad scientific interest. The incorporation of the revised text addressing the concerns and additional data in the paper and supplement markedly improves the manuscript.

There is one labeling issue that needs to be addressed. Line 102 refers the NMY-2 folding in Fig. 1b, but in the Figure panel c is labeled (I believe correctly) as NMY-2. Accordingly line 114 refers to Fig. 1c for MHC-B and that is panel b in the Figure 1. The legend for the Figure 1 (line 756) has these two panels incorrectly identified. This confusion needs attention.

The figure panels were exchanged to fit to the text and legend.